# PROMPT-TUNING LATENT DIFFUSION MODELS FOR INVERSE PROBLEMS

## ABSTRACT

We propose a new method for solving imaging inverse problems using text-to-image latent diffusion models as general priors. Existing methods using latent diffusion models for inverse problems typically rely on simple null text prompts, which can lead to suboptimal performance. To address this limitation, we introduce a method for prompt tuning, which jointly optimizes the text embedding on-the-fly while running the reverse diffusion process. This allows us to generate images that are more faithful to the diffusion prior. In addition, we propose a method to jointly utilize the VAE generative prior. Synergistically leveraging both the diffusion and the VAE prior to solving inverse problems helps to reduce image artifacts, a major problem when using latent diffusion models instead of pixel-based diffusion models. Our combined method, called P2L, outperforms both image- and latent-diffusion model-based inverse problem solvers on a variety of tasks, such as super-resolution, deblurring, and inpainting.

## 1 INTRODUCTION

Imaging inverse problems are often solved by optimizing or sampling a functional that combines a data-fidelity/likelihood term with a regularization term or signal prior (Romano et al., 2017; Venkatakrishnan et al., 2013; Ongie et al., 2020; Kamilov et al., 2023; Kawar et al., 2022; Kadkhodaie & Simoncelli, 2021; Chung et al., 2023b). A common regularization strategy is to use pre-trained image priors from generative models, such as GANs (Bora et al., 2017), VAEs (Bora et al., 2017; González et al., 2022), Normalizing flows (Whang et al., 2021) or Diffusion models (Song et al., 2022; Chung & Ye, 2022).

In particular, diffusion models have gained significant attention as implicit generative priors for solving inverse problems in imaging (Kadkhodaie & Simoncelli, 2021; Whang et al., 2022; Daras et al., 2022; Kawar et al., 2022; Feng et al., 2023; Laroche et al., 2023; Chung et al., 2023b). Leaving the pre-trained diffusion prior intact, one can guide the inference process to perform posterior sampling conditioned on the measurement at inference time by resorting to Bayesian inference. In the end, the ultimate goal of Diffusion model-based Inverse problem Solvers (DIS) would be to act as a fully general inverse problem solver, which can be used not only regardless of the imaging model, but also regardless of the data distribution.

Solving inverse problems in a fully general domain is hard. This directly stems from the difficulty of generative modeling a wide distribution, where it is known that one has to trade-off diversity with fidelity by some means of sharpening the distribution (Brock et al., 2018; Dhariwal & Nichol, 2021). The standard approach in modern diffusion models is to condition on text prompts (Rombach et al., 2022; Saharia et al., 2022b), among them the most popular being Stable Diffusion (SD), a latent diffusion model (LDM), which is itself an under-explored topic in the context of inverse problem solving. While text conditioning is now considered standard practice in content creation including images (Ramesh et al., 2022; Saharia et al., 2022b), 3D (Poole et al., 2023; Wang et al., 2023d), video (Ho et al., 2022), personalization (Gal et al., 2022), and editing (Hertz et al., 2022), it has been completely disregarded in the inverse problem solving context. This is natural, as it is highly ambiguous which text would be beneficial to use when all we have is a degraded measurement. The wrong prompt could easily lead to degraded performance.

In this work, we aim to bridge this gap by proposing a way to *automatically* find the right prompt to condition diffusion models when solving inverse problems. This can be achieved through optimizing

| Prompt | FFHQ | | | | | | ImageNet | | | | | |
|---|---|---|---|---|---|---|---|---|---|---|---|---|
| | SR×8 | | | Inpaint ($p = 0.8$) | | | SR×8 | | | Inpaint ($p = 0.8$) | | |
| | FID↓ | LPIPS↓ | PSNR↑ | FID↓ | LPIPS↓ | PSNR↑ | FID↓ | LPIPS↓ | PSNR↑ | FID↓ | LPIPS↓ | PSNR↑ |
| `""` | 61.16 | 0.327 | 26.49 | 52.34 | 0.241 | **29.78** | 78.68 | 0.397 | 23.49 | 70.87 | 0.350 | 26.20 |
| `"A high quality photo"` | 61.17 | 0.327 | 26.57 | 52.82 | 0.237 | 29.70 | 77.00 | 0.396 | 23.51 | 69.10 | 0.350 | 26.26 |
| `"A high quality photo of a cat"` | 69.03 | 0.377 | 26.39 | 55.15 | 0.248 | 29.63 | 76.69 | 0.402 | **23.63** | 68.48 | 0.355 | 26.13 |
| `"A high quality photo of a dog"` | 66.55 | 0.371 | 26.48 | 55.91 | 0.249 | 29.65 | 76.45 | 0.394 | 23.58 | 67.75 | 0.354 | 26.10 |
| `"A high quality photo of a face"` | 60.41 | 0.325 | 26.74 | 52.33 | 0.239 | 29.69 | 77.32 | 0.403 | 23.60 | 68.83 | 0.352 | 26.20 |
| Proposed | **58.73** | **0.317** | 26.68 | **51.40** | **0.233** | 29.69 | **66.96** | **0.386** | 23.57 | 66.82 | **0.314** | **26.29** |
| PALI prompts from $\boldsymbol{y}$ | 61.33 | 0.329 | **26.81** | 54.34 | 0.249 | 29.76 | 68.28 | 0.388 | 23.57 | 69.55 | 0.355 | 26.26 |
| PALI prompts from $\boldsymbol{x}$ | 60.73 | 0.322 | 26.76 | 52.06 | 0.238 | 29.75 | **66.55** | 0.387 | 23.57 | **64.00** | 0.348 | 26.17 |

Table 1: Difference in restoration performance using LDPS on SR×8 task with varying text prompts. Proposed: text embedding optimized without access to ground truth. PALI prompts from $\boldsymbol{x}/\boldsymbol{y}$: captions are generated with PALI (Chen et al., 2022) from $\boldsymbol{x}$: ground truth clean images / $\boldsymbol{y}$: degraded images. The former can be considered an empirical upper bound.

the continuous text embedding *on-the-fly* while running DIS. We formulate this into a Bayesian framework of updating the text embedding and the latent in an alternating fashion, such that they become gradually aligned during the sampling process. Orthogonal and complementary to embedding optimization, we devise a simple LDM-based DIS (LDIS) that controls the evolution of the latents to stay on the natural data manifold and additionally utilizes the VAE prior for stability of the solutions. We name the algorithm that combines these components P2L, short for **P**rompt-tuning **P**rojected **L**atent diffusion model-based inverse problem solver. In reaching for the ultimate goal of DIS, we focus on 1) **LDM-based DIS** (LDIS) for solving inverse problems in the 2) **fully general domain** (using a single pre-trained checkpoint) that targets 3) **512×512 resolution**[1]. All the aforementioned components are highly challenging, and to the best of our knowledge, have not been studied in conjunction before.

## 2 BACKGROUND

### 2.1 LATENT DIFFUSION MODELS

Diffusion models are generative models that learn to reverse the forward noising process (Sohl-Dickstein et al., 2015; Ho et al., 2020; Song et al., 2021), starting from the initial distribution $p_0(\boldsymbol{x})$, $\boldsymbol{x} \in \mathbb{R}^n$ and approaching the standard Gaussian $p_T(\boldsymbol{x}) = \mathcal{N}(\boldsymbol{0}, \boldsymbol{I})$ as $T \to \infty$ by the forward Gaussian perturbation kernels $p(\boldsymbol{x}_t|\boldsymbol{x}_0) = \mathcal{N}(\boldsymbol{x}_0, t^2\boldsymbol{I})$[2]. The forward/reverse processes can be characterized with Ito stochastic differential equations (SDE). Sampling from the distribution can either be done through solving the reverse SDE, or equivalently by solving the probability-flow ordinary differential equation (PF-ODE) (Song et al., 2021; Karras et al., 2022):

$$d\boldsymbol{x}_t = -t\nabla_{\boldsymbol{x}_t} \log p(\boldsymbol{x}_t)\,dt = \frac{\boldsymbol{x}_t - \mathbb{E}[\boldsymbol{x}_0|\boldsymbol{x}_t]}{t}\,dt,\ \boldsymbol{x}_T \sim p_T(\boldsymbol{x}_T), \tag{1}$$

where we use the Tweedie's formula (Efron, 2011) given as $\mathbb{E}[\boldsymbol{x}_0|\boldsymbol{x}_t] = \boldsymbol{x}_t + t^2\nabla_{\boldsymbol{x}_t} \log p(\boldsymbol{x}_t)$. Here $\nabla_{\boldsymbol{x}_t} \log p_t(\boldsymbol{x}_t)$ is typically approximated with a score network $\boldsymbol{s}_{\boldsymbol{\theta}}(\cdot)$ or a noise estimation network $\boldsymbol{\epsilon}_{\boldsymbol{\theta}}(\cdot)$, and learned through denoising score matching (DSM) (Vincent, 2011) or epsilon-matching loss (Ho et al., 2020).

Image diffusion models that operate on the pixel space $\boldsymbol{x}$ are compute-heavy. One workaround for compute-efficient generative modeling is to leverage a variational autoencoder that maximizes the evidence lower bound (ELBO) (Rombach et al., 2022; Kingma & Welling, 2013). This leads to the following encoder and decoder representation for all $\boldsymbol{x} \sim p_{\text{data}}(\boldsymbol{x}) \in \mathbb{R}^n$:

$$\boldsymbol{x} = \mathcal{D}_{\boldsymbol{\varphi}}(\boldsymbol{z}), \quad \text{where} \quad \boldsymbol{z} = \mathcal{E}_{\boldsymbol{\phi}}(\boldsymbol{x}) := \mathcal{E}_{\boldsymbol{\phi}}^{\boldsymbol{\mu}}(\boldsymbol{x}) + \mathcal{E}_{\boldsymbol{\phi}}^{\boldsymbol{\sigma}}(\boldsymbol{x}) \odot \boldsymbol{\epsilon}, \quad \boldsymbol{\epsilon} \sim \mathcal{N}(0, \boldsymbol{I}), \tag{2}$$

---

[1] All prior works on DIS/LDIS focused on 256×256 resolution. Most LDIS focused their evaluation on a constrained dataset such as FFHQ, and did not scale their method to more general domains such as ImageNet.

[2] Here, we use the choice used in Karras et al. (2022) for simplicity, but use variance preserving (VP) models (Song et al., 2021) for experiments as pre-trained models are available in this form. The different choices can be considered equivalent (Kawar et al., 2022)

where $\mathcal{E}_{\phi}^{\mu}, \mathcal{E}_{\phi}^{\sigma}$ are parts of the encoder that outputs the mean and the variance of the encoder distribution, $\mathcal{D}_{\varphi}$ is the decoder, and $z \in \mathbb{R}^k$ with $k < n$ corresponds to the *latent* representation. After encoding into the latent space (Rombach et al., 2022), one can train a diffusion model in the low-dimensional latent space. Latent diffusion models (LDM) are beneficial in that the computation is cheaper as it operates in a lower-dimensional space, consequently being more suitable for modeling higher dimensional data (e.g. large images of size $\geq 512^2$). The effectiveness of LDMs have democratized the use of diffusion models as the de facto standard of generative models especially for images under the name of Stable Diffusion (SD), which we focus on extensively in this work.

One notable difference of SD from standard image diffusion models (Dhariwal & Nichol, 2021) is the use of text conditioning $\epsilon_{\theta}(\cdot, \mathcal{C})$, where $\mathcal{C}$ is the continuous embedding vector usually obtained through the CLIP text embedder (Radford et al., 2021). As the model is trained with LAION-5B (Schuhmann et al., 2022), a large-scale dataset containing image-text pairs, SD can be conditioned during the inference time to generate images that are aligned with the given text prompt by directly using $\epsilon_{\theta}(\cdot, \mathcal{C})$, or by means of classifier-free guidance (CFG) (Ho & Salimans, 2021).

### 2.2 SOLVING INVERSE PROBLEM WITH (LATENT) DIFFUSION MODELS

Given access to some measurement

$$\boldsymbol{y} = \boldsymbol{A}\boldsymbol{x} + \boldsymbol{n}, \quad \boldsymbol{x} \in \mathbb{R}^n, \ \boldsymbol{y} \in \mathbb{R}^m, \ \boldsymbol{A} \in \mathbb{R}^{m \times n}, \ \boldsymbol{n} \sim \mathcal{N}(\boldsymbol{0}, \sigma_y^2 \boldsymbol{I}_m) \tag{3}$$

where $\boldsymbol{A}$ is the forward operator and $\boldsymbol{n}$ is additive white Gaussian noise, the task is retrieving $\boldsymbol{x}$ from $\boldsymbol{y}$. As the problem is ill-posed, a natural way to solve it is to perform posterior sampling $\boldsymbol{x} \sim p(\boldsymbol{x}|\boldsymbol{y})$ by defining a suitable prior $p(\boldsymbol{x})$. In DIS, diffusion models (i.e. denoisers) act as the implicit prior with the use of the score function.

Earlier methods utilized an alternating projection approach, where hard measurement constraints are applied in-between the denoising steps whether in pixel space (Kadkhodaie & Simoncelli, 2021; Song et al., 2021) or measurement space (Song et al., 2022; Chung & Ye, 2022). Distinctively, projection in the spectral space via singular value decomposition (SVD) to incorporate measurement noise has been developed (Kawar et al., 2021; 2022). Subsequently, methods that aim to approximate the gradient of the log posterior in the diffusion model context have been proposed (Chung et al., 2023b; Song et al., 2023b), expanding the applicability to nonlinear problems. Broadening the range even further, methods that aim to solve blind (Chung et al., 2023a; Murata et al., 2023), 3D (Chung et al., 2023d; Lee et al., 2023), and unlimited resolution problems (Wang et al., 2023b) were introduced. More recently, methods leveraging diffusion score functions within variational inference to solve inverse imaging has been proposed (Mardani et al., 2023; Feng et al., 2023). Notably, all the aforementioned methods utilize *image-domain* diffusion models. Orthogonal to this direction, some of the recent works have shifted their attention to using *latent* diffusion models (Rout et al., 2023; Song et al., 2023a; He et al., 2023), a direction that we follow in this work.

In fact, inverse solvers can be directly linked to posterior sampling from $p(\boldsymbol{x}_0|\boldsymbol{y})$, which can be achieved by modifying Eq. (1) with

$$d\boldsymbol{x}_t = -t \nabla_{\boldsymbol{x}_t} \log p(\boldsymbol{x}_t|\boldsymbol{y}) \, dt = \frac{\boldsymbol{x}_t - \mathbb{E}[\boldsymbol{x}_0|\boldsymbol{x}_t, \boldsymbol{y}]}{t} \, dt, \ \boldsymbol{x}_T \sim p_T(\boldsymbol{x}_T). \tag{4}$$

Here, $\log p(\boldsymbol{x}_t|\boldsymbol{y}) = \log p(\boldsymbol{x}_t) + \log p(\boldsymbol{y}|\boldsymbol{x}_t)$, and the second equality is given by conditioning the Tweedie's formula with $\boldsymbol{y}$, i.e. $\mathbb{E}[\boldsymbol{x}_0|\boldsymbol{x}_t, \boldsymbol{y}] = \boldsymbol{x}_t + t^2 \nabla_{\boldsymbol{x}_t} \log p(\boldsymbol{x}_t|\boldsymbol{y})$. However, as $\log p(\boldsymbol{y}|\boldsymbol{x}_t)$ is intractable, DPS (Chung et al., 2023b) proposes to approximate it with $\log p(\boldsymbol{y}|\boldsymbol{x}_t) \simeq \log p(\boldsymbol{y}|\mathbb{E}[\boldsymbol{x}_0|\boldsymbol{x}_t])$, whose approximation error can be quantified and bounded by the Jensen gap. This idea was recently extended to LDMs in a few recent works (Rout et al., 2023; He et al., 2023)

$$\nabla_{\boldsymbol{z}_t} \log p(\boldsymbol{y}|\boldsymbol{z}_t) \simeq \nabla_{\boldsymbol{z}_t} \log p(\boldsymbol{y}|\mathcal{D}_{\varphi}(\mathbb{E}[\boldsymbol{z}_0|\boldsymbol{z}_t])) = \nabla_{\boldsymbol{z}_t} \|\boldsymbol{y} - \mathcal{D}_{\varphi}(\hat{\boldsymbol{z}}_0)\|_2^2 / \sigma_y^2, \tag{5}$$

with $\hat{\boldsymbol{z}}_0 := \mathbb{E}[\boldsymbol{z}_0|\boldsymbol{z}_t]$. We refer to the sampler that uses the approximation in Eq. (5) as Latent DPS (LDPS) henceforth. Rout et al. (2023) extends LDPS with an additional regularization term by showing that there exists a step size for which guiding the latents towards a fixed point is optimal. He et al. (2023) extends LDPS by using history updates as in Adam (Kingma & Ba, 2015). However, *all* of the existing works in the literature that aim for LDIS, to the best of our knowledge, neglect the use of text embedding by resorting to the use of null text embedding $\mathcal{C}_{\varnothing}$.

---

**Algorithm 1** P2L

---

    **Require:** $\boldsymbol{\epsilon}_{\boldsymbol{\theta}^*}, \boldsymbol{z}_T, \boldsymbol{y}, \mathcal{C}, T, K, \gamma, \lambda_{\mathcal{D}}$

  1:  **for** $t = T$ **to** $1$ **do**

① $\mathcal{C}$ update      2:     $\mathcal{C}_t^* \leftarrow \text{OPTIMIZEEMB}(\boldsymbol{z}_t, \boldsymbol{y}, \mathcal{C}_t^0, K)$          ▷ See Algorithm 2

  3:     $\hat{\boldsymbol{\epsilon}}_t \leftarrow \boldsymbol{\epsilon}_{\boldsymbol{\theta}^*}(\boldsymbol{z}_t, \mathcal{C}_t^*)$

  4:     $\hat{\boldsymbol{z}}_{0|t} \leftarrow (\boldsymbol{z}_t - \sqrt{1 - \bar{\alpha}_t}\hat{\boldsymbol{\epsilon}}_t)/\sqrt{\bar{\alpha}_t}$          ▷ Tweedie's formula

  5:     **if** $(t \mod \gamma) = 0$ **then**

② projection      6:         $\hat{\boldsymbol{x}}_0 \leftarrow \arg\min_{\boldsymbol{x}_0} \|\boldsymbol{y} - \boldsymbol{A}\boldsymbol{x}_0\|_2^2 + \lambda \|\boldsymbol{x}_0 - \mathcal{D}_{\boldsymbol{\varphi}}(\hat{\boldsymbol{z}}_{0|t})\|_2^2$

  7:         $\hat{\boldsymbol{z}}_{0|t} \leftarrow \mathcal{E}_{\boldsymbol{\phi}}(\hat{\boldsymbol{x}}_0)$

  8:     **end if**

③ $\boldsymbol{z}_t$ update      9:     $\boldsymbol{z}'_{t-1} \leftarrow \sqrt{\bar{\alpha}_{t-1}}\hat{\boldsymbol{z}}_{0|t} + \sqrt{1 - \bar{\alpha}_{t-1}}\hat{\boldsymbol{\epsilon}}_t$

 10:     $\boldsymbol{z}_{t-1} \leftarrow \boldsymbol{z}'_{t-1} - \rho_t \nabla_{\boldsymbol{z}_t} \|\boldsymbol{A}\mathcal{D}_{\boldsymbol{\varphi}}(\hat{\boldsymbol{z}}_{0|t}) - \boldsymbol{y}\|$

 11:     $\mathcal{C}_{t-1}^{(0)} \leftarrow \mathcal{C}_t^*$

 12:  **end for**

 13:  **return** $\boldsymbol{x}_0 \leftarrow \mathcal{D}_{\boldsymbol{\varphi}}(\boldsymbol{z}_0)$

---

## 2.3 PROMPT TUNING

In modern language models and vision-langauge models, *prompting* is a standard technique (Radford et al., 2021; Brown et al., 2020) to guide the large pre-trained models to solve downstream tasks. As it has been found that even slight variations in the prompting technique can lead to vastly different outcomes (Kojima et al., 2022), prompt tuning (learning) has been introduced (Shin et al., 2020; Zhou et al., 2022), which defines a *learnable* context vector to optimize over. It was shown that by only optimizing over the continuous embedding vector while maintaining the model parameters fixed, one can achieve a significant performance gain.

In the context of diffusion models, prompt tuning has been adopted for personalization (Gal et al., 2022), where one defines a special token to embed a specific concept with only a few images. Moreover, it has also been demonstrated that one can achieve superior editing performance by optimizing for the null text prompt $\mathcal{C}_{\varnothing}$ (Mokady et al., 2023) before the reverse diffusion sampling process.

## 3 MAIN CONTRIBUTION: THE P2L ALGORITHM

### 3.1 PROMPT-TUNING INVERSE PROBLEM SOLVER

The objective of solving inverse problems is to provide a restoration that is as close as possible to the ground truth given the measurement, whether we are targeting to minimize the distortion or to maximize the perceptual quality (Blau & Michaeli, 2018; Delbracio & Milanfar, 2023). In the context of LDIS,

$$\arg\min_{\boldsymbol{x}, \boldsymbol{c}} \mathcal{L}(\boldsymbol{x}, \boldsymbol{c}) \equiv \arg\min_{\boldsymbol{z}, \boldsymbol{c}} \mathcal{L}(\mathcal{D}_{\boldsymbol{\varphi}}(\boldsymbol{z}), \boldsymbol{c}) \tag{6}$$

where the first equation follows from $\boldsymbol{x} = \mathcal{D}_{\boldsymbol{\varphi}}(\hat{\boldsymbol{z}})$ in deterministic decoder mapping VAE, where $\boldsymbol{c}$ is the text embedding and the loss $\mathcal{L}$ will be explained in more detail in subsequent session. It is easy to see that

$$\arg\min_{\boldsymbol{z}, \boldsymbol{c}} \mathcal{L}(\mathcal{D}_{\boldsymbol{\varphi}}(\boldsymbol{z}), \boldsymbol{c}) \leq \arg\min_{\boldsymbol{z}} \mathcal{L}(\mathcal{D}_{\boldsymbol{\varphi}}(\boldsymbol{z}), \boldsymbol{c} = \mathcal{C}_{\varnothing}), \tag{7}$$

where $\mathcal{C}_{\varnothing}$ is the text embedding from the null text prompt. Notably, by keeping one of the variables fixed, we are optimizing for the *upper bound* of the objective that we truly wish to optimize over. It would be naturally beneficial to optimize the LHS of Eq. (7), rather than the RHS used in the previous methods.

**A motivating example** To see Eq. (7) in effect, we conduct two canonical experiments with 256 test images of FFHQ (Karras et al., 2019) and ImageNet (Deng et al., 2009): super-resolution (SR) of scale ×8 and inpainting with 80% of the pixels randomly dropped, using the LDPS algorithm. Keeping all the other hyper-parameters fixed, we only vary the text condition for the diffusion model.

In addition to using a general text prompt, we use PALI (Chen et al., 2022) to provide captions from the ground truth images ($\boldsymbol{x}$) and from the measurements ($\boldsymbol{y}$) and use them when running LDPS. Further details on the experiment can be found in Appendix B. In Table 1, we first see that simply varying the text prompts can lead to dramatic difference the performance. For instance, we see an increase of over 10 FID when we use the text prompts from PALI for the task of $\times 8$ SR on ImageNet. In contrast, using the prompts generated from $\boldsymbol{y}$ often degrades the performance (e.g. inpainting) as the correct captions cannot be generated. From this motivating example, it is evident that additionally optimizing for $\boldsymbol{c}$ would bring us gains that are orthogonal to the development of the solvers (Rout et al., 2023; He et al., 2023; Song et al., 2023a), a direction that has not been explored in the literature. Indeed, from the table, we see that by applying our prompt tuning approach, we achieve a large performance gain, sometimes even outperforming the PALI captions which has full access to the ground truth when attaining the text embeddings.

**Overall framework**    To effectively utilize the Latent Diffusion Inverse Solver (LDIS), it is essential to ensure two key criteria: 1) consistency with the measurements, and 2) the feasibility of the solution as per the latent diffusion model. In the context of prompt-tuning LDIS, the optimization of the prompt is also a crucial aspect. Instead of trying to meet these objectives all at once, our approach in this paper is a more pragmatic one, where we alternate between these optimization goals. Specifically, we focus on maintaining data consistency and optimizing the prompt in every iteration. This approach often intertwines with the need to uphold the feasibility condition dictated by latent diffusion. We will delve into the specifics of this methodology in the subsequent section.

## 3.2    DATA CONSISTENCY AND PROMPT TUNING

Existing LDIS approaches attempt to sample from $p(\boldsymbol{x}_0|\boldsymbol{y}, \mathcal{C}_\varnothing)$, as it is hard to specify a generally good condition $\mathcal{C}$ when all we have access to is the corrupted $\boldsymbol{y}$. Hence, our goal is to find a good $\mathcal{C}$ *on-the-fly* while solving for the inverse problem. Before diving into the design of the algorithm, let us first revisit Eq. (4) for the case where we consider $\mathcal{C}$ as a condition.

$$d\boldsymbol{z}_t = -t\nabla_{\boldsymbol{z}_t} \log p(\boldsymbol{z}_t|\boldsymbol{y}, \mathcal{C})\, dt = \frac{\boldsymbol{z}_t - \mathbb{E}[\boldsymbol{z}_0|\boldsymbol{z}_t, \boldsymbol{y}, \mathcal{C}]}{t}\, dt, \tag{8}$$

where $\mathbb{E}[\boldsymbol{z}_0|\boldsymbol{z}_t, \boldsymbol{y}, \mathcal{C}]$ is approximated with the *empirical* conditional posterior mean and

$$\nabla_{\boldsymbol{z}_t} \log p(\boldsymbol{z}_t|\boldsymbol{y}, \mathcal{C}_t) = \nabla_{\boldsymbol{z}_t} \log p(\boldsymbol{z}_t|\mathcal{C}_t) + \nabla_{\boldsymbol{z}_t} \log p(\boldsymbol{y}|\boldsymbol{z}_t, \mathcal{C}_t) \tag{9}$$

$$\simeq \boldsymbol{s}_{\boldsymbol{\theta}^*}(\boldsymbol{z}_t, \mathcal{C}_t) + \rho_t \nabla_{\boldsymbol{z}_t} \|\boldsymbol{y} - \boldsymbol{A}\mathcal{D}_{\boldsymbol{\varphi}}(\mathbb{E}[\boldsymbol{z}_0|\boldsymbol{z}_t, \mathcal{C}_t])\|_2^2, \tag{10}$$

where we used the neural network parameterized score function $\boldsymbol{s}_{\boldsymbol{\theta}^*}(\boldsymbol{z}_t, \mathcal{C}_t) \simeq \nabla_{\boldsymbol{z}_t} \log p(\boldsymbol{z}_t|\mathcal{C}_t)^3$, and the second term comes from DPS (Chung et al., 2023b) by defining the prompt conditioned posterior mean $\hat{\boldsymbol{z}}_0^{(\mathcal{C})} := \mathbb{E}[\boldsymbol{z}_0|\boldsymbol{z}_t, \mathcal{C}]$.

$$p(\boldsymbol{y}|\boldsymbol{z}_t, \mathcal{C}) = p(\boldsymbol{y}|\mathcal{D}_{\boldsymbol{\varphi}}(\boldsymbol{z}_t), \mathcal{C}) \overset{\text{(DPS)}}{\simeq} p(\boldsymbol{y}|\mathcal{D}_{\boldsymbol{\varphi}}(\hat{\boldsymbol{z}}_0^{(\mathcal{C})})). \tag{11}$$

Equipped with the approximation in Eq. (11), we propose a sampler that alternates between the optimization of $\mathcal{C}$ while keeping $\boldsymbol{z}_t$ fixed, the sampling of $\boldsymbol{z}_t$ while keeping the $\mathcal{C}$ fixed from the previous iteration.

**Step ①: $\mathcal{C}$ update**    As $\mathcal{C}$ is an unknown, we propose to optimize it such that the data fidelity is met. Specifically, from Eq. (13), we use the following optimization:

$$\mathcal{C}_t^* = \arg\min_{\mathcal{C}} \|\boldsymbol{y} - \boldsymbol{A}\mathcal{D}_{\boldsymbol{\varphi}}(\mathbb{E}[\boldsymbol{z}_0|\boldsymbol{z}_t, \boldsymbol{y}, \mathcal{C}])\|_2^2. \tag{12}$$

This corresponds to the OPTIMIZEEMB in Algorithm 1, with details of the optimization function in Algorithm 2. Further details can be found in Appendix A,D.

**Step ③: $\boldsymbol{z}_t$ update**    Plugging in the obtained $\mathcal{C}_t^*$ into Eq. (13), we can obtain the standard LDPS gradient equipped with $\mathcal{C}_t^*$, the optimized text embedding for step $t$. i.e.,

$$\nabla_{\boldsymbol{z}_t} \log p(\boldsymbol{z}_t|\boldsymbol{y}, \mathcal{C}_t^*) \simeq \boldsymbol{s}_{\boldsymbol{\theta}^*}(\boldsymbol{z}_t, \mathcal{C}_t^*) + \rho_t \nabla_{\boldsymbol{z}_t} \|\boldsymbol{y} - \boldsymbol{A}\mathcal{D}_{\boldsymbol{\varphi}}(\mathbb{E}[\boldsymbol{z}_0|\boldsymbol{z}_t, \mathcal{C}_t^*])\|_2^2, \tag{13}$$

---

$^3$Only using Eq. (13) with $\mathcal{C}_t = \mathcal{C}_\varnothing$ would result in standard LDPS.

where we set $\rho_t$ to be the step size that weights the likelihood, similar to Chung et al. (2023b). We summarize our alternating sampling method in Algorithm 1 and Algorithm 2, based on DDIM sampling, with standard noise schedule notations adopted from Ho et al. (2020). The reason we refer this step to Step ③ in Algorithm 1 is that this is intertwined with the feasibility constraint from the latent diffusion model at every $\gamma$ iteration, which will be described in the following.

## 3.3 ENFORCING FEASIBILITY

**Step ②: projection**  To obtain the feasible solutions by the latent diffusion model, we additionally incorporate the VAE prior. Specifically, we consider the following loss, which is the maximum a posteriori (MAP) objective under the VAE prior in Eq. (2) with isotropic covariance.

$$\mathcal{L}(\boldsymbol{x}, \boldsymbol{z}) = \|\boldsymbol{y} - \boldsymbol{A}\mathcal{D}_{\boldsymbol{\varphi}}(\boldsymbol{z})\|_2^2 + \zeta\|\boldsymbol{z} - \mathcal{E}_{\boldsymbol{\mu}}(\boldsymbol{x})\|_2^2, \tag{14}$$

where $\zeta$ absorbs the weighting caused by the variance of respective terms. We can solve Eq. (14) by splitting the variables similar in spirit to the alternating direction method of multipliers (Boyd et al., 2011). Namely, using the decoder approximation and setting $\boldsymbol{x} = \mathcal{D}_{\boldsymbol{\varphi}}(\boldsymbol{z})$, the optimization problem with respect to $\boldsymbol{x}$ becomes

$$\min_{\boldsymbol{x}} \|\boldsymbol{y} - \boldsymbol{A}\boldsymbol{x}\|_2^2 + \zeta\|\boldsymbol{z} - \mathcal{E}_{\boldsymbol{\phi}}^{\boldsymbol{\mu}}(\mathcal{D}_{\boldsymbol{\varphi}}(\boldsymbol{z}))\|_2^2 + \lambda\|\boldsymbol{x} - \mathcal{D}_{\boldsymbol{\varphi}}(\boldsymbol{z}) + \boldsymbol{\eta}\|_2^2. \tag{15}$$

Here, we set dual variable $\boldsymbol{\eta}$ as a zero vector and do not consider its update. This leads to

$$\boldsymbol{x}^* = \arg\min_{\boldsymbol{x}} \|\boldsymbol{y} - \boldsymbol{A}\boldsymbol{x}\|_2^2 + \lambda\|\boldsymbol{x} - \mathcal{D}_{\boldsymbol{\varphi}}(\boldsymbol{z})\|_2^2, \tag{16}$$

which can be solved with negligible computation cost such as conjugate gradient (CG). Subsequently, using the encoder approximation and setting $\boldsymbol{z} = \mathcal{E}_{\boldsymbol{\phi}}(\boldsymbol{x})$ with $\boldsymbol{\eta} = \boldsymbol{0}$, the optimization problem with respect to $\boldsymbol{z}$ reads

$$\boldsymbol{z}^* = \arg\min_{\boldsymbol{z}} \|\boldsymbol{y} - \boldsymbol{A}\mathcal{D}_{\boldsymbol{\varphi}}\mathcal{E}_{\boldsymbol{\phi}}(\boldsymbol{x})\|_2^2 + \zeta\|\boldsymbol{z} - \mathcal{E}_{\boldsymbol{\phi}}(\boldsymbol{x})\|_2^2, \tag{17}$$

which has a closed-form solution $\boldsymbol{z}_0^* = \mathcal{E}_{\boldsymbol{\phi}}(\boldsymbol{x}_0)$. Solving for Eq. (14) is performed on the *clean* data manifold with the latents obtained through the Tweedie's formula, similar to Chung et al. (2023c); Zhu et al. (2023), as presented in line 6-7 of Algorithm 1.

In practice, we choose to apply Eq. (16),Eq. (17) every few iteration to control dramatic changes in the sampling, and to save computation. Nevertheless, solving Eq. (16) requires access to $\boldsymbol{A}^\top$, which is often non-trivial to define. Contrarily, our jax implementation enables defining $\boldsymbol{A}^\top$ through jax.vjp. For further discussion, see Appendix E. Note that by Eq. (17), we guarantee that the clean latents stay on the *range space* of the encoder[4], which minimizes the train-test time discrepancy. This is natural as the training of LDMs is done with latents that are in the range space of $\mathcal{E}_{\boldsymbol{\phi}}$. For this reason, we often denote the method proposed in this section simply as "projection".

## 3.4 FURTHER DISCUSSIONS

**Relation to the previous approach**  The crucial component that delineates LDM is the existence of VAE. When naively using the VAE, the decoder introduces a significant amount of error especially when the estimated clean latent $\hat{\boldsymbol{z}}_0^{(\mathcal{C})}$ falls off the manifold of the clean latents, which inevitably happens with the LDPS approximation. Rout et al. (2023) proposed Posterior Sampling using Latent Diffusion (PSLD) to regularize the update steps on the latent so that the clean latents are led to the fixed point of the successive application of decoding-encoding. Formally, omitting the dependence on $\mathcal{C}$, they use the following gradient step

$$\nabla_{\boldsymbol{z}_t} \log(\boldsymbol{y}|\boldsymbol{z}_t) \simeq \nabla_{\boldsymbol{z}_t}\left(\|\boldsymbol{y} - \boldsymbol{A}\mathcal{D}_{\boldsymbol{\varphi}}(\hat{\boldsymbol{z}}_0)\|_2^2 + \lambda\|\hat{\boldsymbol{z}}_0 - \mathcal{E}_{\boldsymbol{\phi}}(\mathcal{D}_{\boldsymbol{\varphi}}(\hat{\boldsymbol{z}}_0))\|_2^2\right), \tag{18}$$

where the additional regularization term weighted by $\lambda$ leads $\hat{\boldsymbol{z}}_0$ towards the fixed point. From our context summarized in the loss function in Eq. (14), their approach is equivalent to using the following loss function:

$$\mathcal{L}_{PSLD}(\boldsymbol{z}) = \mathcal{L}(\mathcal{D}_{\boldsymbol{\varphi}}(\boldsymbol{z}), \boldsymbol{z}) \tag{19}$$

---

[4]This is different from the range-null space decomposition of the imaging operator, as proposed in Wang et al. (2023c).

In order word, rather than using a soft constraint by alternating minimization between $\boldsymbol{x}$ and $z$ as in our approach, they enforce hard constraint between the $\boldsymbol{x}$ and $\boldsymbol{z}$ in the form of $\boldsymbol{x} = \mathcal{D}_{\boldsymbol{\varphi}}(\boldsymbol{z})$. Although this provided a first successful demonstration of the LDIS, the nature of VAE is not used in the formulation, which may explain why our approach is outperforming.

**Targetting arbitrary resolution** Despite its fully convolutional nature, as SD was trained with $64{\times}64$ latents ($\leftrightarrow 512 \times 512$ images), the performance degrades when we aim to deal with larger dimensions, again due to train-test time discrepancy. Several works aimed to mitigate this issue by processing the latents with strided patches (Bar-Tal et al., 2023; Jiménez, 2023; Wang et al., 2023a) that increases the computational burden by roughly $\mathcal{O}(n^2)$. In contrast, we show that our projection approach, used *without* any patch processing, can outperform previous methods that rely on patches, resulting in significantly faster inference speed. In Appendix F, we show that our approach is also useful for targeting arbitrary resolution image restoration, as the errors accumulated by processing latents in higher dimensions can be corrected through our projection approach.

## 4 EXPERIMENTS

**Datasets, Models** We consider two different well-established datasets: 1) FFHQ 512×512 (Karras et al., 2019), and 2) ImageNet 512×512 (Deng et al., 2009). For the former, we use the first 1000 images for testing, similar to Chung et al. (2023b). For the latter, we choose 1k images out of 10k test images provided in Saharia et al. (2022a) by interleaved sampling, i.e. using images of index 0, 10, 20, etc. after ordering by name. For the latent diffusion model, we choose SD v1.4 pre-trained on the LAION dataset for all the experiments, including the baseline comparison methods based on LDM. As there is no publicly available image diffusion model that is trained on an identical dataset, we choose ADM (Dhariwal & Nichol, 2021) trained on ImageNet 512×512 data as the universal prior when implementing baseline image-domain DIS. Note that this discrepancy may lead to an unfair advantage in the performance for evaulation on ImageNet, and an unfair disadvantage in the performance when evaluating on FFHQ. All experiments were done on NVIDIA A100 40GB GPUs.

**Inverse Problems** We test our method on the following degradations: 1) Super-resolution from ×8 averagepooling, 2) Inpainting from 10-20% free-form masking as used in Saharia et al. (2022a), 3) Gaussian deblurring from an image convolved with a $61{\times}61$ size Gaussian kernel with $\sigma = 3.0$, 4) Motion deblurring from an image convolved with a $61{\times}61$ motion kernel that is randomly sampled with intensity $0.5^5$, following Chung et al. (2023b). For all degradations, we include mild additive white Gaussian noise with $\sigma_y = 0.01$.

**Evaluation** As the main objective of this study is to improve the performance of LDIS, we mainly focus our evaluation on the comparison against the current SOTA LDIS: we compare against LDPS, GML-DPS (Rout et al., 2023), PSLD (Rout et al., 2023), and LDIR (He et al., 2023). Notably, all LDIS including the proposed P2L use 1000 NFE DDIM sampling with $\eta = 0.0^6$. We additionally compare against SOTA image-domain DIS: DPS (Chung et al., 2023b), Diff-PIR (Zhu et al., 2023), DDS (Chung et al., 2023c), and ΠGDM (Song et al., 2023b). For DPS, we use 1000 NFE DDIM sampling. For Diff-PIR, DDS, and ΠGDM, we use 100 NFE DDIM sampling. We choose the optimal $\eta$ values for these algorithms through grid-search. Details about the comparison methods can be found in Appendix D.3. We perform a quantitative evaluation with standard metrics: PSNR, FID, and LPIPS.

**Comparison against baseline** In all of the inverse problems that we consider in the paper, our method outperforms all the baselines by quite a large margin in terms of perceptual quality, measured by FID and LPIPS, while keeping the distortion at a comparable level against the current state-of-the-art methods. Especially, we see about 10 FID decrease in deblurring and inpainting tasks compared to the runner up in both FFHQ and ImageNet dataset (See Tables 8,2). The superiority can also be clearly seen in Fig. 1, where P2L achieves stable, high-quality reconstruction throughout all tasks. Results from both LDPS and PSLD often contain local grid-like artifacts (Red boxes in Figures) and are blurry. With P2L, the restored images are sharpened while the artifacts are effectively removed.

---

[5]https://github.com/LeviBorodenko/motionblur

[6]The parameter $\eta$ indicates the stochasticity of the sampler. $\eta = 0.0$ leads to deterministic PF-ODE.

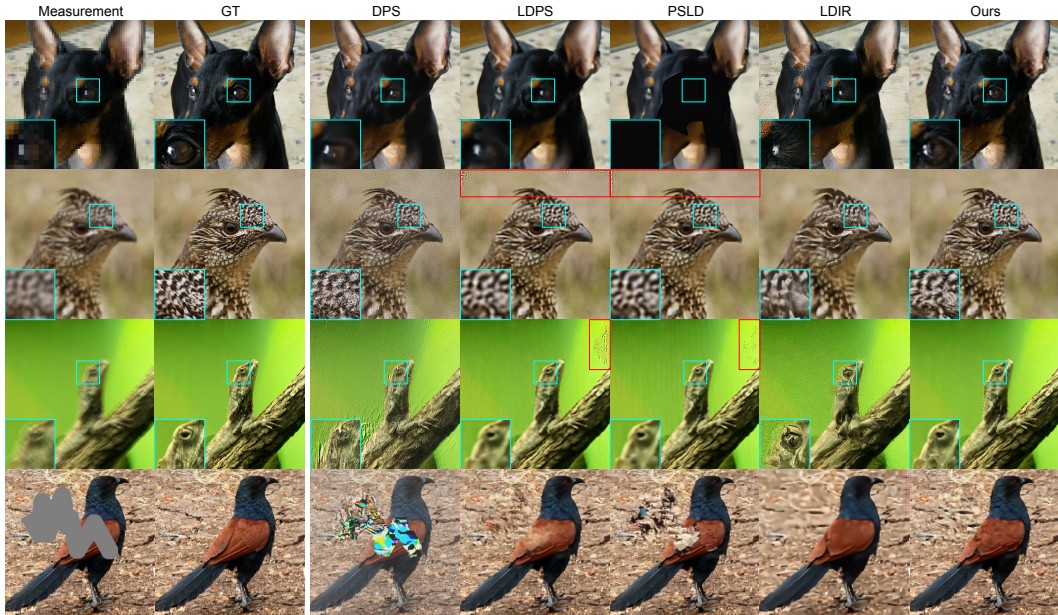

Figure 1: Inverse problem solving results on ImageNet $512 \times 512$ test set. Row 1: SR$\times 8$, Row 2: gaussian deblurring, Row 3: motion deblurring, row 4: inpainting.

| Method | SR ($\times 8$) | | | Deblur (motion) | | | Deblur (gauss) | | | Inpaint | | |
|---|---|---|---|---|---|---|---|---|---|---|---|---|
| | FID↓ | LPIPS↓ | PSNR↑ | FID↓ | LPIPS↓ | PSNR↑ | FID↓ | LPIPS↓ | PSNR↑ | FID↓ | LPIPS↓ | PSNR↑ |
| P2L (ours) | **51.81** | **0.386** | **23.38** | **54.11** | **0.360** | **24.79** | **39.10** | **0.325** | 25.11 | **32.82** | **0.229** | 21.99 |
| LDPS | 61.09 | 0.475 | 23.21 | 71.12 | 0.441 | 23.32 | 48.17 | 0.392 | 24.91 | 46.72 | 0.332 | 21.54 |
| GML-DPS (Rout et al., 2023) | 60.36 | 0.456 | 23.21 | 59.08 | 0.403 | 24.35 | 45.33 | 0.377 | **25.44** | 47.30 | 0.294 | 21.12 |
| PSLD (Rout et al., 2023) | 60.81 | 0.471 | 23.17 | 59.63 | 0.398 | 24.21 | 45.44 | 0.376 | 25.42 | 40.57 | 0.251 | 20.92 |
| LDIR (He et al., 2023) | 63.46 | 0.480 | 22.23 | 88.51 | 0.475 | 21.37 | 72.10 | 0.506 | 22.45 | 50.65 | 0.313 | **23.28** |
| DDS (Chung et al., 2023c) | 203.2 | 1.213 | 12.72 | 84.67 | 0.925 | 14.52 | 70.51 | 0.835 | 16.58 | 60.18 | 0.354 | 17.03 |
| DPS (Chung et al., 2023b) | 54.61 | 0.544 | 20.70 | 71.99 | 0.599 | 19.62 | 98.33 | 0.910 | 15.05 | 71.70 | 0.360 | 15.15 |
| DiffPIR (Zhu et al., 2023) | 488.3 | 1.182 | 13.44 | 87.04 | 0.622 | 19.32 | 79.31 | 0.755 | 20.55 | 45.97 | 0.300 | 20.11 |
| ΠGDM (Song et al., 2023b) | 53.00 | 0.490 | 21.08 | 75.35 | 0.682 | 18.66 | 70.26 | 0.797 | 21.96 | 65.75 | 0.322 | 16.84 |

Table 2: Quantitative evaluation (PSNR, LPIPS, FID) of inverse problem solving on ImageNet $512 \times 512$-1k validation dataset. **Bold**: best, underline: second best. Methods that are not LDM-based are shaded in gray.

LDIR are less prone to artifacts owing to the smoothed history gradient updates, but often results in unrealistic textures and deviations from the measurement, which is also reflected in having the lowest PSNR among the LDIS-class methods. In contrast, P2L is free from such drawbacks even when leveraging Adam-like gradient update steps. However, it should be noted that the compute time for P2L linearly increases as we increase the number of training iterations for the text embedding. The compute time for $K = 0$ is similar to other LDIS baselines, but it becomes slower if $K$ becomes larger. Devising a more time-efficient way to perform text embedding optimization is thus a promising future research direction. For further details on the runtime analysis, see Appendix C.

One rather surprising finding is the heavy downgrade in the performance for DIS methods. Even on in-distribution ImageNet test data, methods such as DPS and DiffPIR become very unstable. This can be attributed to the generative prior being poor: directly training diffusion models on high-resolution images often result in poor performance[7]. This observation again points to the importance of developing methods that can leverage foundation models when aiming for general domain higher-resolution data. See Appendix G for further results. As a final note, we believe that the compromise in PSNR is related to the imperfectness of the VAE used in SD v1.4[8], and we expect

---

[7]Consequently, for $\geq 512 \times 512$ resolution, either using latent diffusion or using cascaded models (Saharia et al., 2022b) are popular.

[8]Auto-encoding 1000 ground-truth test images result in the following metrics: FFHQ (PSNR): $29.66 \pm 2.29$, ImageNet (PSNR): $27.12 \pm 4.38$.

| Design components | | | FFHQ | | | | ImageNet | | | |
|---|---|---|---|---|---|---|---|---|---|---|
| | | | SR×8 | | Inpaint ($p=0.8$) | | SR×8 | | Inpaint ($p=0.8$) | |
| Projection | $\mathbf{\Gamma}$ | Prompt tuning | FID↓ | PSNR↑ | FID↓ | PSNR↑ | FID↓ | PSNR↑ | FID↓ | PSNR↑ |
| ✗ | ✗ | ✗ | 61.16 | 26.49 | 52.34 | **29.78** | 78.68 | 23.49 | 70.87 | 26.20 |
| ✗ | ✗ | ✓ | 58.73 | **26.68** | 51.40 | 29.69 | 76.40 | **23.52** | 67.06 | 26.32 |
| ✓ | ✗ | ✗ | 55.91 | 26.37 | 48.71 | 29.68 | 74.22 | 23.16 | 66.92 | 26.08 |
| ✓ | ✓ | ✗ | 55.68 | 26.43 | 47.76 | 29.70 | 74.01 | 23.32 | 65.45 | 26.29 |
| ✓ | ✓ | ✓ | **52.96** | 26.64 | **46.92** | 29.63 | **70.08** | 23.48 | **59.26** | 26.12 |

Table 3: Ablation studies on the design components

| $\sigma_y$ | $\mathbf{\Gamma}$ | PSNR | FID |
|---|---|---|---|
| 0.0 | glue | 26.51 | 54.69 |
| | Ours | **26.80** | **54.58** |
| 0.01 | glue | 26.39 | 56.47 |
| | Ours | **26.43** | **55.68** |
| 0.05 | glue | 23.86 | 68.99 |
| | Ours | **24.92** | **65.90** |

Table 4: Choice of $\mathbf{\Gamma}$

such degradation to be mitigated when switching to better, larger autoencoders such as SDXL (Podell et al., 2023).

**Design components** In Table 3, we perform an ablation study on the design components of the proposed method. From the table, we confirm that prompt tuning, projection to the range space of the encoder, and performing proximal update step (denoted as $\mathbf{\Gamma}$) before the projection all contributes to the gain in the performance. It is important that these gains are synergistic, and one component does not hamper the other. In the Appendix Tab. 7, we further show that our prompt-tuning approach is robust to the variation in the hyper-parameters (learning rate, number of iterations). Specifically, among the 9 configurations that we try, only the one with 5 iterations, lr=0.001 is inferior to not using prompt tuning. In Fig. 2, we visualize the progress of $\mathcal{D}(\hat{\boldsymbol{z}}_0)$ through time $t$ starting from the same random seed, comparing LDPS, PSLD, and LDPS + projection (row 4 of Tab. 6). Here, we see that our proposed projection approach effectively suppresses the artifacts that arise during the reconstruction process, whereas PSLD introduces additional artifacts. Furthermore, in Appendix F, we show that our approach is also useful for targetting arbitrary resolution image restoration, as the errors accumulated by processing latents in higher dimensions can be corrected through our projection approach. Remarkably, we see that our approach often offers better results (e.g. see Fig. 4) than operating in strided patches (Bar-Tal et al., 2023; Jiménez, 2023), which requires quadratic scaling of compute time.

**Choice of $\mathbf{\Gamma}$** When projecting to the range space of $\mathcal{E}$, we choose to use the proximal optimization strategy in Eq. (16). Instead, one could resort to projection to the measurement subspace ("gluing" of Rout et al. (2023)) by using $\mathbf{\Gamma}(\hat{\boldsymbol{x}}_0) = \boldsymbol{A}^\top \boldsymbol{y} + (\boldsymbol{I} - \boldsymbol{A}^\top \boldsymbol{A})\hat{\boldsymbol{x}}_0$. In Table 4, we compare our choice of $\mathbf{\Gamma}$ against the gluing on various noise levels on FFHQ SR×8. We see that for all noise levels, the proximal steps consistently outperform the gluing, even when $\mathbf{\Gamma}$ is applied every $\gamma = 4$ steps of reverse diffusion. Furthermore, due to the noise-amplifying nature of projection, the differences become more pronounced as we increase the noise level. The difference in the compute time between the two choices is minimal: 331.7 [s] vs 333.2 [s] measured in wall-clock time using RTX 3090 GPU per the restoration of a single image when we compare gluing vs. proximal optimization.

## 5 CONCLUSION

We proposed P2L, a latent diffusion model-based inverse problem solver that introduces two new strategies. First, a prompt tuning method to optimize the continuous input text embedding used for diffusion models was developed. We observed that our strategy can boost the performance by a good margin compared to the usage of null text embedding that prior works employ. Second, a projection approach to keep the latents in the range space of the encoder during the reverse diffusion process was proposed. We show that our approach paves way to jointly utilizing diffusion generative prior and the VAE generative prior. Our approach effectively mitigated the artifacts that often arise during inverse problem solving, while also sharpening the final output. P2L outperforms previous diffusion model-based inverse problem solvers that operate on the latent and the image domain.

**Limitations** While prompt tuning enhances the performance, it also incurs additional computational complexity as additional forward/backward passes through the latent diffusion model and the decoder is necessary. Consequently, the method will need future investigations when aiming for time-critical applications. As we optimize the continuous text embeddings rather than the discrete text directly, it is hard to decipher what the text embedding after the optimization has converged to. This is a limitation of the text embedder used for SD, as CLIP does not utilize a decoder. We could instead opt for the use

of Imagen (Saharia et al., 2022b), where T-5 with an encoder-decoder architecture is used, where one could easily check the learned text from our prompt-tuning scheme. Moreover, we did not consider the usage of CFG, which would enable flexible control on the degree of sharpening. Extending the prompt tuning idea to jointly optimize the embedding of the conditional and the unconditional model may be an interesting direction of future research. Although we have provided a unified optimization perspective to derive our algorithm, we did not perform convergence analysis in this work. Further analysis of P2L would be an interesting direction of future research, similar to what was done in Rout et al. (2023).

**Ethics statement**  While our method can lead to advancements in areas such as computational imaging, medical imaging, and other fields where inverse problems are prevalent, we also recognize the potential for misuse in areas like deepfake generation or unauthorized data reconstruction, naturally leading from the use of generative models. The potential bias within the training dataset of the diffusion model may be potentially amplified with the usage of our method. We have taken care to ensure that our experiments adhere to ethical guidelines, using publicly available datasets or those for which we have obtained explicit permissions. We urge the community to adopt responsible practices when applying our findings and to consider the broader societal implications of the technology.

**Reproducibility statement**  In order to facilitate reproducibility, We detail our implementation in the form of Algorithms (Alg. 1,2,3), and pseudo-code (Fig. 3). The specific hyper-parameters chosen for the method is detailed in Appendix D.

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

## A BACKGROUND ON DIFFUSION MODELS

**Lemma 1** (Tweedie's formula). *Given a Gaussian perturbation kernel* $p(\boldsymbol{x}_t|\boldsymbol{x}_0) = \mathcal{N}(\boldsymbol{x}_t; s_t\boldsymbol{x}_0, \sigma_t^2\boldsymbol{I})$, *the posterior mean is given by*

$$\mathbb{E}[\boldsymbol{x}_0|\boldsymbol{x}_t] = \frac{1}{\alpha_t}(\boldsymbol{x}_t + \sigma_t^2\nabla_{\boldsymbol{x}_t}\log p(\boldsymbol{x}_t)) \tag{20}$$

*Proof.*

$$\nabla_{\boldsymbol{x}_t}\log p(\boldsymbol{x}_t) = \frac{\nabla_{\boldsymbol{x}_t}p(\boldsymbol{x}_t)}{p(\boldsymbol{x}_t)} \tag{21}$$

$$= \frac{1}{p(\boldsymbol{x}_t)}\nabla_{\boldsymbol{x}_t}\int p(\boldsymbol{x}_t|\boldsymbol{x}_0)p(\boldsymbol{x}_0)\,d\boldsymbol{x}_0 \tag{22}$$

$$= \frac{1}{p(\boldsymbol{x}_t)}\int \nabla_{\boldsymbol{x}_t}p(\boldsymbol{x}_t|\boldsymbol{x}_0)p(\boldsymbol{x}_0)\,d\boldsymbol{x}_0 \tag{23}$$

$$= \frac{1}{p(\boldsymbol{x}_t)}\int p(\boldsymbol{x}_t|\boldsymbol{x}_0)\nabla_{\boldsymbol{x}_t}\log p(\boldsymbol{x}_t|\boldsymbol{x}_0)p(\boldsymbol{x}_0)\,d\boldsymbol{x}_0 \tag{24}$$

$$= \int p(\boldsymbol{x}_0|\boldsymbol{x}_t)\nabla_{\boldsymbol{x}_t}\log p(\boldsymbol{x}_t|\boldsymbol{x}_0)\,d\boldsymbol{x}_0 \tag{25}$$

$$= \int p(\boldsymbol{x}_0|\boldsymbol{x}_t)\frac{s_t\boldsymbol{x}_0 - \boldsymbol{x}_t}{\sigma_t^2}\,d\boldsymbol{x}_0 \tag{26}$$

$$= \frac{s_t\mathbb{E}[\boldsymbol{x}_0|\boldsymbol{x}_t] - \boldsymbol{x}_t}{\sigma_t^2}. \tag{27}$$

Rearranging the terms, we achieve the conclusion. □

Lemma 1 lets us compute the posterior mean when we have access to the score function. In diffusion models, we parametrize the score function with a neural network and train it through denoising score matching

$$\theta^* = \arg\min_\theta \mathbb{E}_{t\sim U[0,1],\boldsymbol{x}_0\sim p_{\text{data}},\boldsymbol{\epsilon}\sim\mathcal{N}(0,\boldsymbol{I})}\|\boldsymbol{s}_\theta(\boldsymbol{x}_t,t) - \nabla_{\boldsymbol{x}_t}\log p(\boldsymbol{x}_t|\boldsymbol{x}_0)\|_2^2. \tag{28}$$

Let us consider the case of DDPM (Ho et al., 2020) with the forward perturbation kernel $p(\boldsymbol{x}_t|\boldsymbol{x}_0) = \mathcal{N}(\boldsymbol{x}_t; \sqrt{\bar{\alpha}_t}\boldsymbol{x}_0, (1-\bar{\alpha}_t)\boldsymbol{I})$[9]. Then, we have the following alternative parametrizations

$$\boldsymbol{s}_{\theta^*}(\boldsymbol{x}_t,t) = -\frac{1}{\sqrt{1-\bar{\alpha}_t}}\boldsymbol{\epsilon}_{\theta^*}(\boldsymbol{x}_t,t) = \frac{\sqrt{\bar{\alpha}_t}D_{\theta^*}(\boldsymbol{x}_0) - \boldsymbol{x}_t}{\sqrt{1-\bar{\alpha}_t}}, \tag{29}$$

where the second parametrization comes from epsilon-matching (Ho et al., 2020) and is mostly used throughout the work, and the last parametrization directly estimates the posterior mean by regarding the diffusion model as a denoiser.

**Corollary 1** (Conditional Tweedie's formula).

$$\mathbb{E}[\boldsymbol{x}_0|\boldsymbol{x}_t,\boldsymbol{y}] = \frac{1}{s_t}(\boldsymbol{x}_t + \sigma_t^2\nabla_{\boldsymbol{x}_t}\log p(\boldsymbol{x}_t|\boldsymbol{y})) \tag{30}$$

The corollary is a simple consequence of conditioning the Tweedie's formula with an additional variable $\boldsymbol{y}$. As $\log p(\boldsymbol{x}_t|\boldsymbol{y})$ is intractable, we can estimate Eq. (30), with the choices of $s_t, \sigma_t$ made from DDPM, with Chung et al. (2023b)

$$\mathbb{E}[\boldsymbol{x}_0|\boldsymbol{x}_t,\boldsymbol{y}] = \frac{1}{\sqrt{\bar{\alpha}_t}}(\boldsymbol{x}_t + (1-\bar{\alpha}_t)\nabla_{\boldsymbol{x}_t}(\log p(\boldsymbol{x}_t) + \log p(\boldsymbol{y}|\boldsymbol{x}_t))) \tag{31}$$

$$\stackrel{\text{(DPS)}}{\approx} \frac{1}{\sqrt{\bar{\alpha}_t}}(\boldsymbol{x}_t + (1-\bar{\alpha}_t)(\boldsymbol{s}_{\theta^*}(\boldsymbol{x}_t,t) + \nabla_{\boldsymbol{x}_t}\log p(\boldsymbol{y}|\hat{\boldsymbol{x}}_0))) \tag{32}$$

$$= \hat{\boldsymbol{x}}_0 + \frac{1-\bar{\alpha}_t}{\sqrt{\bar{\alpha}_t}}\nabla_{\boldsymbol{x}_t}\log p(\boldsymbol{y}|\hat{\boldsymbol{x}}_0) \tag{33}$$

---

[9]In the discrete setup, $\bar{\alpha}_t := \prod_{i=1}^t \alpha_t$, and $\alpha_t := 1 - \beta_t$ with $q(\boldsymbol{x}_t|\boldsymbol{x}_{t-1}) = \mathcal{N}(\boldsymbol{x}_t; \sqrt{1-\beta_t}\boldsymbol{x}_{t-1}, \beta_t\boldsymbol{I})$

where $\hat{\boldsymbol{x}}_0 := D_{\theta^*}(\boldsymbol{x}_t, t)$. Further, we can circumvent the need to backpropagate through the diffusion model and save computation by using the DDS approximation (Chung et al., 2023c)

$$\mathbb{E}[\boldsymbol{x}_0|\boldsymbol{x}_t, \boldsymbol{y}] \overset{\text{(DDS)}}{\approx} \hat{\boldsymbol{x}}_0 + \frac{1 - \bar{\alpha}_t}{\sqrt{\bar{\alpha}_t}} \nabla_{\hat{\boldsymbol{x}}_0} \log p(\boldsymbol{y}|\hat{\boldsymbol{x}}_0), \tag{34}$$

where the difference stems from that we take the gradient w.r.t. $\hat{\boldsymbol{x}}_0$ rather than $\boldsymbol{x}_t$. Running Eq. (4) with the approximations Eq. (32) or Eq. (34) amounts to approximately sampling from the posterior distribution.

## B    PROOF-OF-CONCEPT EXPERIMENT

For the caption generation with PALI, we simply take the captions with the highest score. Examples of the captions generated from PALI are presented in Fig. 8. In our initial experiments, we found that using PALI captions directly did not directly lead to an improvement in the performance, as it only describes the *content* of the image, and says nothing about the *quality* of the image. Therefore, we use the following text prompts for the oracle "A high quality photo of a {PALI_prompt}", similar to the general text prompts.

For both inverse problems (SR×8, inpainting with $p = 0.8$), we use the LDPS algorithm with 1000 NFE and $\eta = 0.0$. We apply prompt tuning algorithm per denoising step as indicated in Algorithm 2, with $K = 5$ and learning rate of $1e - 4$. When optimizing for the text embedding, we initialize it with the embedding vector from the token "A high quality photo of a face" for FFHQ, and "A high quality photo" for ImageNet in the case of inpainting. Note that for the latter, we did not find much performance difference when initializing from the null text prompt, or even initializing it with "A high quality photo of a dog". For ×8 SR, we initialize the text embeddings from PALI captions generated from $\boldsymbol{y}$, as we empirically observe that PALI captions from $\boldsymbol{y}$ still have a relatively good coarse description about the given image.

## C    RUNTIME ANALYSIS

In Tab. 5, we include the runtime for each algorithm used in the paper when solving inverse problems with diffusion models, measured in wall-clock time [s] with a single RTX 3090 GPU. Note that P2L $(K = 0)$ corresponds to the case where we do not use prompt-tuning, and only apply the idea of leveraging the VAE prior (i.e. encoder range space projection). In this case, the compute time is roughly equivalent to the LDIS baselines. As we increase the number of iterations for prompt embedding optimization, the required computation time approximately linearly increases. In this regard, P2L requires more compute against other LDIS baselines as we additionally optimize for the text prompt, which can be considered a downside of the approach. However, it should be noted that P2L is the first approach that shows the possibility and feasibility of the approach. While it may not be computationally efficient at this point, P2L would be a good cornerstone that future works can build upon to devise faster, more efficient solvers.

| Method | Time [s] | Type |
|---|---|---|
| P2L $(K = 5)$ | 1982.7 | |
| P2L $(K = 3)$ | 1333.6 | |
| P2L $(K = 1)$ | 657.3 | |
| P2L $(K = 0)$ | 333.2 | Latent |
| LDPS | 313.9 | diffusion |
| GML-DPS (Rout et al., 2023) | 390.6 | |
| PSLD (Rout et al., 2023) | 408.7 | |
| LDIR (He et al., 2023) | 317.2 | |
| DDS (Chung et al., 2023c) | 20.1 | |
| DPS (Chung et al., 2023b) | 291.0 | Pixel |
| DiffPIR (Zhu et al., 2023) | 21.2 | diffusion |
| ΠGDM (Song et al., 2023b) | 30.2 | |

Table 5: Comparison in compute time for each method using RTX 3090 GPU in wall-clock time [s].

## D    IMPLEMENTATION DETAILS

### D.1    $\mathcal{C}$ UPDATE PROMPT TUNING

We consider the following optimization problem

$$\mathcal{C}^* = \arg\min_{\mathcal{C}} \|\boldsymbol{y} - \boldsymbol{A}\mathcal{D}\left(\mathbb{E}[\boldsymbol{z}_0|\boldsymbol{z}_t, \boldsymbol{y}, \mathcal{C}]\right)\|_2^2, \tag{35}$$

| | FFHQ | | | | ImageNet | | | |
|---|---|---|---|---|---|---|---|---|
| problem | Deblur (motion) | Deblur (gauss) | SR×8 | inpaint | Deblur (motion) | Deblur (gauss) | SR×8 | inpaint |
| Gradient type | Adam | Adam | GD | Adam | Adam | GD | GD | GD |
| $\rho_t$ | 0.05 | 0.05 | 1.0 | 0.05 | 0.1 | $\bar{\alpha}_t$ | $15\bar{\alpha}_t$ | 0.5 |
| $\gamma$ | 5 | 4 | 4 | 3 | 5 | 4 | 4 | 3 |
| $\lambda$ | 1.0 | 1.0 | 1.0 | 0.1 | 1.0 | 1.0 | 1.0 | 0.1 |
| $K$ | 3 | 5 | 5 | 1 | 3 | 3 | 3 | 1 |
| learning rate | $5e-5$ | $1e-4$ | $1e-4$ | $1e-4$ | $1e-5$ | $1e-4$ | $1e-5$ | $1e-4$ |

Table 6: Hyper-parameter choice for the proposed method. White shade: hyper-parameters related to gradient updates, blue shade: hyper-parameters related to projecting onto the range space of $\mathcal{E}$, red shade: hyper-parameters related to prompt tuning.

where Eq. (35) is performed for every timestep $t$ during the inference stage. Here, we approximate the conditional posterior mean as

$$\mathbb{E}[\boldsymbol{z}_0|\boldsymbol{z}_t, \boldsymbol{y}, \mathcal{C}] = \frac{1}{\sqrt{\bar{\alpha}_t}}\boldsymbol{z}_t + \frac{1-\bar{\alpha}_t}{\sqrt{\bar{\alpha}_t}}\left(\nabla_{\boldsymbol{z}_t}\log p(\boldsymbol{z}_t|\mathcal{C}) + \nabla_{\boldsymbol{z}_t}\log p(\boldsymbol{y}|\boldsymbol{z}_t, \mathcal{C})\right) \tag{36}$$

$$\simeq \hat{\boldsymbol{z}}_0^{(\mathcal{C})} + \frac{1-\bar{\alpha}_t}{\sqrt{\bar{\alpha}_t}}\nabla_{\boldsymbol{z}_t}\log p(\boldsymbol{y}|\hat{\boldsymbol{z}}_0^{(\mathcal{C})}) \tag{37}$$

$$\simeq \hat{\boldsymbol{z}}_0^{(\mathcal{C})} + \frac{1-\bar{\alpha}_t}{\sqrt{\bar{\alpha}_t}}\nabla_{\hat{\boldsymbol{z}}_0^{(\mathcal{C})}}\log p(\boldsymbol{y}|\hat{\boldsymbol{z}}_0^{(\mathcal{C})}), \tag{38}$$

which is the consequence of the DDS approximation in Eq. (34). Notice that we update our embeddings to improve the fidelity Eq. (35). However, in practice, this also leads to higher quality images in terms of perception. For optimizing Eq. (35), we use Adam with the learning rate and the number of iterations as denoted in Table 6 for every $t$. In practice, we choose a static step size $\rho = 1.0$ with the gradient of the norm, which was shown to be effective in (Chung et al., 2023b). The resulting prompt tuning algorithm is summarized in Algorithm 2.

---

**Algorithm 2** Prompt tuning

1: **function** OPTIMIZEEMB($\boldsymbol{z}_t, \boldsymbol{y}, \mathcal{C}_t^{(0)}, K$)
2:     **for** $k = 1$ **to** $K$ **do**

Approximation of $\mathbb{E}[\boldsymbol{z}_0|\boldsymbol{z}_t, \mathcal{C}]$ $\begin{cases} \\ \\ \\ \end{cases}$

3:         $\hat{\boldsymbol{\epsilon}}_t \leftarrow \boldsymbol{\epsilon}_{\boldsymbol{\theta}^*}(\boldsymbol{z}_t, \mathcal{C}_t^{(k-1)})$
4:         $\hat{\boldsymbol{z}}_{0|t} \leftarrow (\boldsymbol{z}_t - \sqrt{1-\bar{\alpha}_t}\hat{\boldsymbol{\epsilon}}_t)/\sqrt{\bar{\alpha}_t}$
5:         $\hat{\boldsymbol{z}}'_{0|t} \leftarrow \hat{\boldsymbol{z}}_{0|t} - \rho\nabla_{\hat{\boldsymbol{z}}_{0|t}}\|\boldsymbol{y} - \mathcal{D}_{\boldsymbol{\varphi}}(\hat{\boldsymbol{z}}_{0|t})\|$
6:         $\hat{\boldsymbol{x}}_{0|t} \leftarrow \mathcal{D}_{\boldsymbol{\varphi}}(\hat{\boldsymbol{z}}'_{0|t})$

Optimizing Eq. (12). $\begin{cases} \\ \\ \end{cases}$

7:         $\mathcal{L} \leftarrow \|\boldsymbol{A}\hat{\boldsymbol{x}}_{0|t}(\mathcal{C}_t^{(k-1)}) - \boldsymbol{y}\|_2^2$
8:         $\mathcal{C}_t^{(k)} \leftarrow \mathcal{C}_t^{(k-1)} - \mathtt{AdamGrad}(\mathcal{L}_t)$
9:     **end for**
10:    **return** $\mathcal{C}_t^* \leftarrow \mathcal{C}_t^{(K)}$
11: **end function**

---

---

**Algorithm 3** P2L: Adam

---

**Require:** $\epsilon_{\theta^*}, z_T, y, \mathcal{C}, T, K, \gamma, \beta_1, \beta_2, \varepsilon, \Gamma$
1:   $m_T \leftarrow \texttt{np.zeros\_like}(z_T)$
2:   $v_T \leftarrow \texttt{np.zeros\_like}(z_T)$
3:   **for** $t = T$ to $1$ **do**
4:      $\mathcal{C}_t^* \leftarrow \textsc{OptimizeEmb}(z_t, y, \mathcal{C}_t^0, K)$
5:      $\hat{\epsilon}_t \leftarrow \epsilon_{\theta^*}(z_t, \mathcal{C}_t^*)$
6:      $\hat{z}_{0|t} \leftarrow (z_t - \sqrt{1 - \bar{\alpha}_t}\hat{\epsilon}_t)/\sqrt{\bar{\alpha}_t}$
7:      **if** $(t \mod \gamma) = 0$ **then**
8:        $\hat{z}'_{0|t} \leftarrow \mathcal{E}\left(\Gamma\left(\mathcal{D}(\hat{z}_{0|t})\right)\right)$
9:      **end if**
10:     $z'_{t-1} \leftarrow \sqrt{\bar{\alpha}_{t-1}}\hat{z}'_{0|t} + \sqrt{1 - \bar{\alpha}_{t-1}}\hat{\epsilon}_t$
11:     $g \leftarrow \nabla_{z_t}\|\mathcal{A}\mathcal{D}(\hat{z}_{0|t}) - y\|$
12:     $\hat{m}_{t-1} \leftarrow (\beta_1 m_t + (1 - \beta_1)g)/(1 - \beta_1)$
13:     $\hat{v}_{t-1} \leftarrow (\beta_2 v_t + (1 - \beta_2)(g \circ g))/(1 - \beta_2)$
14:     $z_{t-1} \leftarrow z'_{t-1} - \rho_t \frac{\hat{m}_{t-1}}{\sqrt{\hat{v}_{t-1}} + \varepsilon}$
15:     $\mathcal{C}_{t-1}^{(0)} \leftarrow \mathcal{C}_t^*$
16:   **end for**
17:   **return** $x_0 \leftarrow \mathcal{D}(z_0)$

---

### D.2 $z_t$ UPDATE

In Table 6, there are two gradient types: GD and Adam. For GD, we use standard gradient descent steps as presented in Algorithm 1. For Adam, using the same prompt tuning Algorithm 2, we adopt a history gradient update scheme as proposed in He et al. (2023) to arrive at Algorithm 3. Note that the hyper-parameters of the Adam update were fixed to be $\beta_1 = 0.9, \beta_2 = 0.999, \varepsilon = 1e - 8$, which is the default setting. We only search for the optimal step size $\rho_t$ via grid search, which is set to 0.1 for motion deblurring in ImageNet, and 0.05 otherwise.

### D.3 COMPARISON METHODS

**LDPS**   LDPS can be considered a straightforward extension image domain DPS (Chung et al., 2023b). The three works that we review in this section (He et al., 2023; Rout et al., 2023; Song et al., 2023a) all consider LDPS as a baseline. In LDPS, we have the following update scheme

$$z_{t-1} = \text{DDIM}(z_t) - \rho\nabla_{z_t}\|y - \mathcal{A}\mathcal{D}(\hat{z}_0)\|_2, \tag{39}$$

where $\rho$ is the step size, and $\text{DDIM}(\cdot)$ denotes a single step of DDIM sampling. We use a static step size of $\rho = 1$, widely adopted in literature.

**LDIR (He et al., 2023)**   Using Adam-like history gradient update scheme, a single iteration of the algorithm can be summarized as follows

$$g_t = \nabla_{z_t}\|y - \mathcal{A}\mathcal{D}(\hat{z}_0)\| \tag{40}$$

$$\hat{m}_t = (\beta_1 m_{t-1} + (1 - \beta_1)g_t)/(1 - \beta_1) \tag{41}$$

$$\hat{v}_t = (\beta_2 v_{t-1} + (1 - \beta_2)(g_t \circ g_t))/(1 - \beta_2) \tag{42}$$

$$z_{t-1} = \text{DDIM}(z_t) - \rho\frac{\hat{m}_t}{\sqrt{\hat{v}_t} + \varepsilon}, \tag{43}$$

where $\circ$ denotes element-wise product, and $\beta_1, \beta_2, \varepsilon$ are the hyperparameters of the sampling scheme. As LDIR uses a momentum-based update scheme, we have smoother gradient transitions. We fix $\beta_1 = 0.9, \beta_2 = 0.999, \varepsilon = 1e - 8$ to be identical to when using the proposed method. The step size $\rho$ is chosen to be the optimal value found through grid search: 0.1 for ImageNet motion deblurring, and 0.05 otherwise.

**GML-DPS, PSLD (Rout et al., 2023)**   GML-DPS attempts to regularize the predicted clean latent $\hat{z}_0$ to be a fixed point after encoding and decoding. Formally, the update step reads

$$z_{t-1} = \text{DDIM}(z_t) - \rho\nabla_{z_t}\left(\|y - \mathcal{A}\mathcal{D}(\hat{z}_0)\|_2 + \gamma\|\hat{z}_0 - \mathcal{E}(\mathcal{D}(\hat{z}_0))\|_2\right). \tag{44}$$

| steps | 0 | 1 | | | 3 | | | 5 | | |
|---|---|---|---|---|---|---|---|---|---|---|
| lr | - | $1e-5$ | $1e-4$ | $1e-3$ | $1e-5$ | $1e-4$ | $1e-3$ | $1e-5$ | $1e-4$ | $1e-3$ |
| FID | 61.16 | 60.66 | 59.60 | **57.61** | 60.11 | 59.34 | 60.19 | 60.02 | 58.59 | 62.67 |
| PSNR | 26.49 | 26.69 | 26.71 | 26.73 | **26.78** | 26.70 | 26.61 | 26.73 | 26.17 | 26.38 |

Table 7: Robustness to hyper-parameters in prompt-tuning. FFHQ SR$\times$8 on 256 test images. **Bold**: best, underline: second best.

Further, PSLD applies an orthogonal projection onto the subspace of $\boldsymbol{A}$ in between decoding and encoding to enforce fidelity

$$\boldsymbol{z}_{t-1} = \mathrm{DDIM}(\boldsymbol{z}_t) - \rho\nabla_{\boldsymbol{z}_t}\left(\|\boldsymbol{y} - \boldsymbol{A}\mathcal{D}(\hat{\boldsymbol{z}}_0)\|_2 + \gamma\|\hat{\boldsymbol{z}}_0 - \mathcal{E}(\boldsymbol{A}^\top\boldsymbol{y} + (\boldsymbol{I} - \boldsymbol{A}^\top\boldsymbol{A})\mathcal{D}(\hat{\boldsymbol{z}}_0))\|_2\right). \tag{45}$$

We use the static step size of $\rho = 1$, and choose $\gamma = 0.1$, as advised in Rout et al. (2023). GML-DPS and PSLD are closest to the proposed method in spirit, as these methods attempt to guide the latents to stay closer to the natural manifold by enforcing them to be a fixed point after autoencoding. The difference is that these approaches use gradient guidance while we try to explicitly project the latents into the the natural manifold.

**DPS (Chung et al., 2023b)** DPS is a DIS that utilizes the following update scheme[10]

$$\boldsymbol{x}_{t-1} = \mathrm{DDIM}(\boldsymbol{x}_t) - \nabla_{\boldsymbol{x}_t}\left(\|\boldsymbol{y} - \boldsymbol{A}\hat{\boldsymbol{x}}_0\|_2\right). \tag{46}$$

The optimal value of $\eta$ was found through grid search for each inverse problem: $\eta = 0.0$ for SR$\times$8, and $\eta = 1.0$ for others.

**$\Pi$GDM (Song et al., 2023b)** Similar to DPS, $\Pi$GDM considers the following gradient update scheme

$$\boldsymbol{x}_{t-1} = \mathrm{DDIM}(\boldsymbol{x}_t) - \left((\boldsymbol{y} - \boldsymbol{A}\hat{\boldsymbol{x}}_0)^\top(r_t^2\boldsymbol{A}\boldsymbol{A}^\top + \sigma^2\boldsymbol{I})^{-1}\boldsymbol{A}\frac{\partial\hat{\boldsymbol{x}}_0}{\partial\boldsymbol{x}_t}\right)^\top, \tag{47}$$

where $r_t$ is a hyper-parameter and $\sigma$ is the noise level of the measurement. We take $r_t$ as advised in (Song et al., 2023b), and use 100 step DDIM sampling with $\eta = 1.0$ for all experiments.

**DDS (Chung et al., 2023c)** The following updates are used

$$\hat{\boldsymbol{x}}_0' = \arg\min_{\boldsymbol{x}}\frac{1}{2}\|\boldsymbol{y} - \boldsymbol{A}\boldsymbol{x}\|_2^2 + \frac{\gamma}{2}\|\boldsymbol{x} - \hat{\boldsymbol{x}}_0\|_2^2 \tag{48}$$

$$\boldsymbol{x}_{t-1} = \sqrt{\bar{\alpha}_{t-1}}\hat{\boldsymbol{x}}_0' + \sqrt{1 - \bar{\alpha}_{t-1} - \eta^2\tilde{\beta}_{t-1}^2}\hat{\boldsymbol{\epsilon}}_t + \eta\tilde{\beta}_{t-1}\boldsymbol{\epsilon}, \tag{49}$$

where Eq. (48) is solved through CG with 5 iterations, $\gamma = 1.0$. $\eta = 0.0$ is chosen for Gaussian deblurring, and $\eta = 1.0$ for the rest of the inverse problems.

**DiffPIR (Zhu et al., 2023)** Similar to DDS, the following updates are used

$$\hat{\boldsymbol{x}}_0' = \arg\min_{\boldsymbol{x}}\frac{1}{2}\|\boldsymbol{y} - \boldsymbol{A}\boldsymbol{x}\|_2^2 + \frac{\lambda\sigma^2\bar{\alpha}_t}{2(1 - \bar{\alpha}_t)}\|\boldsymbol{x} - \hat{\boldsymbol{x}}_0\|_2^2 \tag{50}$$

$$\boldsymbol{x}_{t-1} = \sqrt{\bar{\alpha}_{t-1}}\hat{\boldsymbol{x}}_0' + \sqrt{1 - \bar{\alpha}_{t-1}}(\sqrt{1 - \zeta}\hat{\boldsymbol{\epsilon}}_t + \sqrt{\zeta}\boldsymbol{\epsilon}), \tag{51}$$

where $\sigma$ is the noise level of the measurement, and $\lambda, \zeta$ are hyper-parameters. Unlike DDS, the solution to Eq. (50) is obtained as a closed-form solution. These hyper-parameters are found through grid search. SR$\times$8: $\zeta = 0.35, \lambda = 35.0$ / Deblur: $\zeta = 0.3, \lambda = 7.0$ / Inpaint: $\zeta = 1.0/\lambda = 7.0$.

## E EFFICIENT IMPLEMENTATION IN JAX

---

[10]The original work only considered DDPM sampling. We consider DDIM as a generalization of DDPM as it can be retrieved with $\eta = 1.0$.

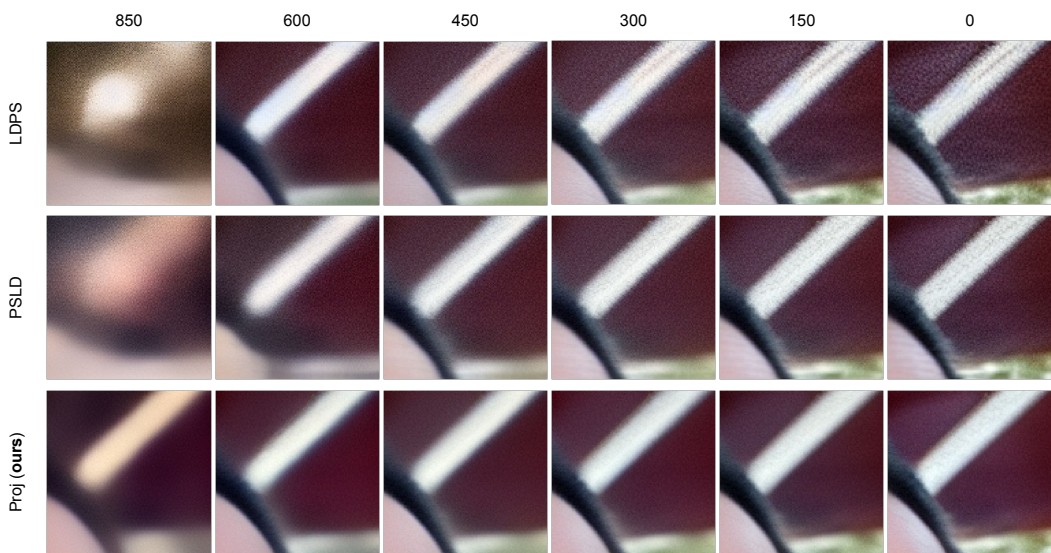

Figure 2: Close-up of the progress of $\mathcal{D}(\hat{z}_0)$ through time $t$ when solving $\times 8$ SR on FFHQ.

```
ones = jnp.ones(x.shape)
_, _AT = jax.vjp(A_funcs.A, ones)
AT = lambda y: _AT(y)[0]
A_funcs.AT = AT
def cg_A(x, cg_lamb):
    return A_funcs.AT(A_funcs.A(x)) + cg_lamb * x
hatx0 = D(hatz0)
cg_y = A_funcs.AT(y) + cg_lamb * hatx0
hatx0, _ = jax.scipy.sparse.linalg.cg(cg_A, cg_y, x0=hatx0)
```

Figure 3: Defining $A^\top$ can be automatically achieved through jax.vjp given that $A$ is differentiable.

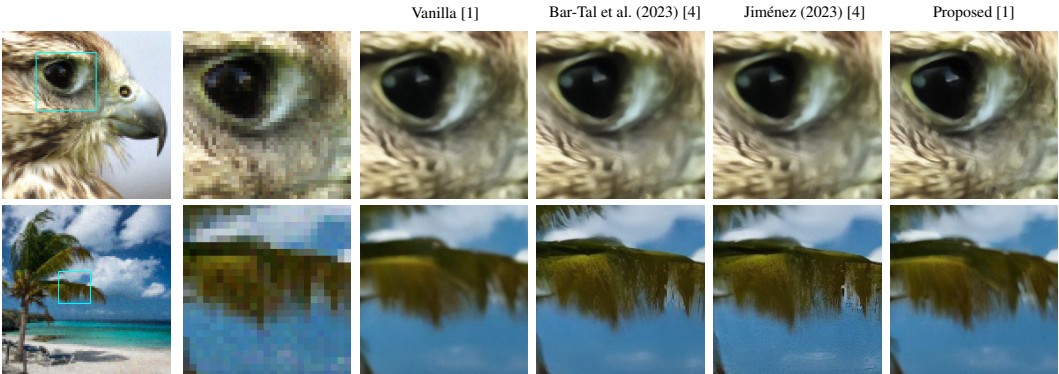

Figure 4: Results on $\times 8$ SR on DIV2K validation set of $768 \times 768$ resolution. [Diffusion NFE per denoising step]. Vanilla and proposed process the latent as a whole.

In model-based inverse problem solving, having access to efficient computation of the adjoint $A^\top$ is a must. Here, we consider a general case of solving linear inverse problems where the computation of SVD is too costly, and hence one has to define the adjoint operator manually (e.g. computed tomography). Furthermore, for cases such as deblurring from circular convolution, one needs to carefully design the operator, as there are many potential pitfalls (e.g. boundary, size mismatch). These are more often than not the limiting factors of the applicability of the model-based approaches for solving inverse problems. We show in Fig. 3 that this can be much alleviated by using jax, as we can implicitly define a transpose operator with reverse-mode automatic differentiation (Baydin et al., 2018). We note this design was also established in (Balke et al., 2022).

## F  TARGETTING ARBITRARY RESOLUTION

For SD, using an encoder to convert from the image to the latent space reduces the dimension by $\times 8$. When training SD, the diffusion model that operates on the latent space was trained with $64\times 64$ latents, obtained from $512\times 512$ images. When the image that we wish to restore (or generate) is larger than $512\times 512$, the latents will also be larger than $64\times 64$. In this case, due to the train-test time discrepancy, the results that we get will be suboptimal if one processes the larger latent as a whole (Fig. 5 (a)). A natural way to counteract this discrepancy is to process the latents in patches[11]. When processing in patches of size $64\times 64$ with stride 32 on both directions, it requires us 4 score function NFEs per denoising step (Fig. 5 (c),(d)). Bar-Tal et al. (2023) uniformly weights the overlapping patches, and Jiménez (2023) weights the patches with Gaussian weights with variance 0.01. The downside of these methods is that the number NFEs required for inference scales quadratically with the size of the image.

Notice that all methods that aim for high-resolution synthesis using latent diffusion models only focus on better dealing with the latents and use the decoding part as-is. This is due to the fact that the diffusion models that act in the latent space is more sensitive to the change in the input resolution, and hence the error could easily accumulate if we operate on larger latents directly. On the other hand, VAE is much more robust to the change in the input resolution. When given a latent that stays within the range space of the encoder, the decoder is able to produce a high-quality image directly even when the input size is larger than $64\times 64$. In this regard, we can project this latent to the range space of $\mathcal{E}$ by setting $\hat{z}'_0 = \mathcal{D}(\mathbf{\Gamma}(\mathcal{E}(\hat{z}_0)))$ for every few steps, as illustrated in Fig. 5 (b). Even though the proposed method is considerably faster than patch-based methods (Bar-Tal et al., 2023; Jiménez, 2023), we see that one can achieve a comparable, or superior performance, as presented in Fig. 4. Furthermore, in Fig. 6, we show that we can use both patching method and the projection method simultaneously, achieving the best results.

(a) Vanilla                                             (b) Proposed

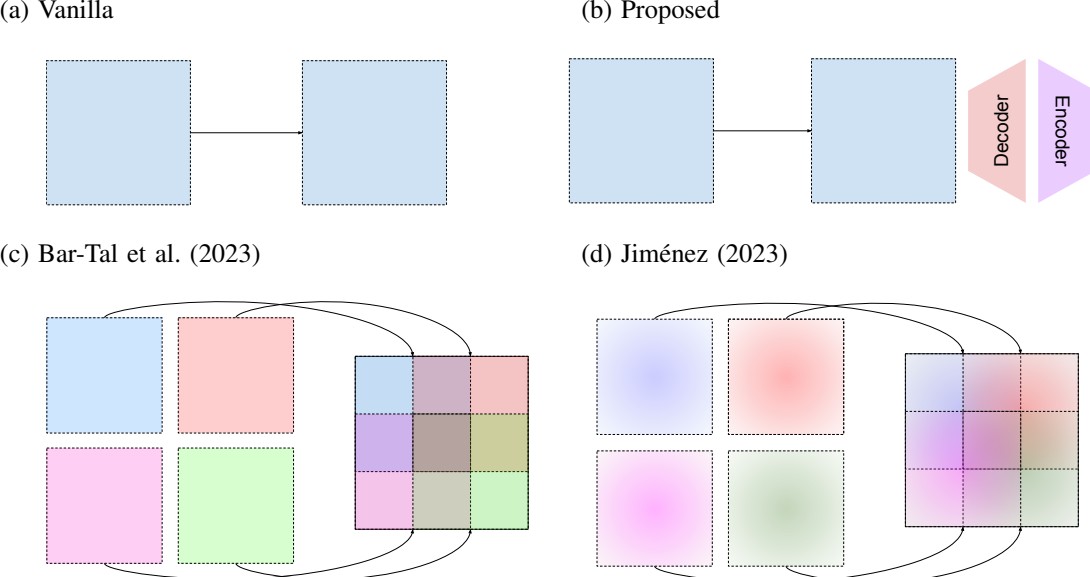

(c) Bar-Tal et al. (2023)                               (d) Jiménez (2023)

Figure 5: Method comparison for processing higher resolution images in the latent space.

## G  FURTHER EXPERIMENTAL RESULTS

---

[11]For all the experiments considered in this paper, we consider $768\times 768$ images ($96\times 96$ latents).

| Method | SR (×8) | | | Deblur (motion) | | | Deblur (gauss) | | | Inpaint | | |
|---|---|---|---|---|---|---|---|---|---|---|---|---|
| | FID↓ | LPIPS↓ | PSNR↑ | FID↓ | LPIPS↓ | PSNR↑ | FID↓ | LPIPS↓ | PSNR↑ | FID↓ | LPIPS↓ | PSNR↑ |
| P2L (ours) | **31.23** | **0.290** | 28.55 | 28.34 | **0.302** | **27.23** | 30.62 | 0.299 | 26.97 | **26.27** | **0.168** | 25.29 |
| LDPS | 36.81 | 0.292 | **28.78** | 58.66 | 0.382 | 26.19 | 45.89 | 0.334 | 27.82 | 46.10 | 0.311 | 23.07 |
| GML-DPS (Rout et al., 2023) | 41.65 | 0.318 | 28.50 | 47.96 | 0.352 | 27.16 | 42.60 | 0.320 | **28.49** | 36.31 | 0.208 | 23.10 |
| PSLD (Rout et al., 2023) | 36.93 | 0.335 | 26.62 | 47.71 | 0.348 | 27.05 | 41.04 | 0.320 | 28.47 | 35.01 | 0.207 | 23.10 |
| LDIR (He et al., 2023) | 36.04 | 0.345 | 25.79 | **24.40** | 0.376 | 24.40 | 35.61 | 0.341 | 25.75 | 37.23 | 0.250 | **25.47** |
| DDS (Chung et al., 2023c) | 262.0 | 1.278 | 13.01 | 88.70 | 1.014 | 14.68 | 74.02 | 0.932 | 17.03 | 113.6 | 0.421 | 17.92 |
| DPS (Chung et al., 2023b) | 47.65 | 0.340 | 21.81 | 65.91 | 0.601 | 21.11 | 100.2 | 0.983 | 15.71 | 137.7 | 0.692 | 15.35 |
| DiffPIR (Zhu et al., 2023) | 141.1 | 1.266 | 13.80 | 72.02 | 0.664 | 21.03 | 69.15 | 0.751 | 22.27 | 33.92 | 0.238 | 24.91 |
| ΠGDM (Song et al., 2023b) | 42.07 | 0.311 | 22.05 | 60.08 | 0.531 | 21.08 | 70.32 | 0.788 | 21.99 | 140.6 | 0.738 | 16.83 |

Table 8: Quantitative evaluation (PSNR, LPIPS, FID) of inverse problem solving on FFHQ 512×512-1k validation dataset. **Bold**: best, underline: second best. Methods that are not LDM-based are shaded in gray.

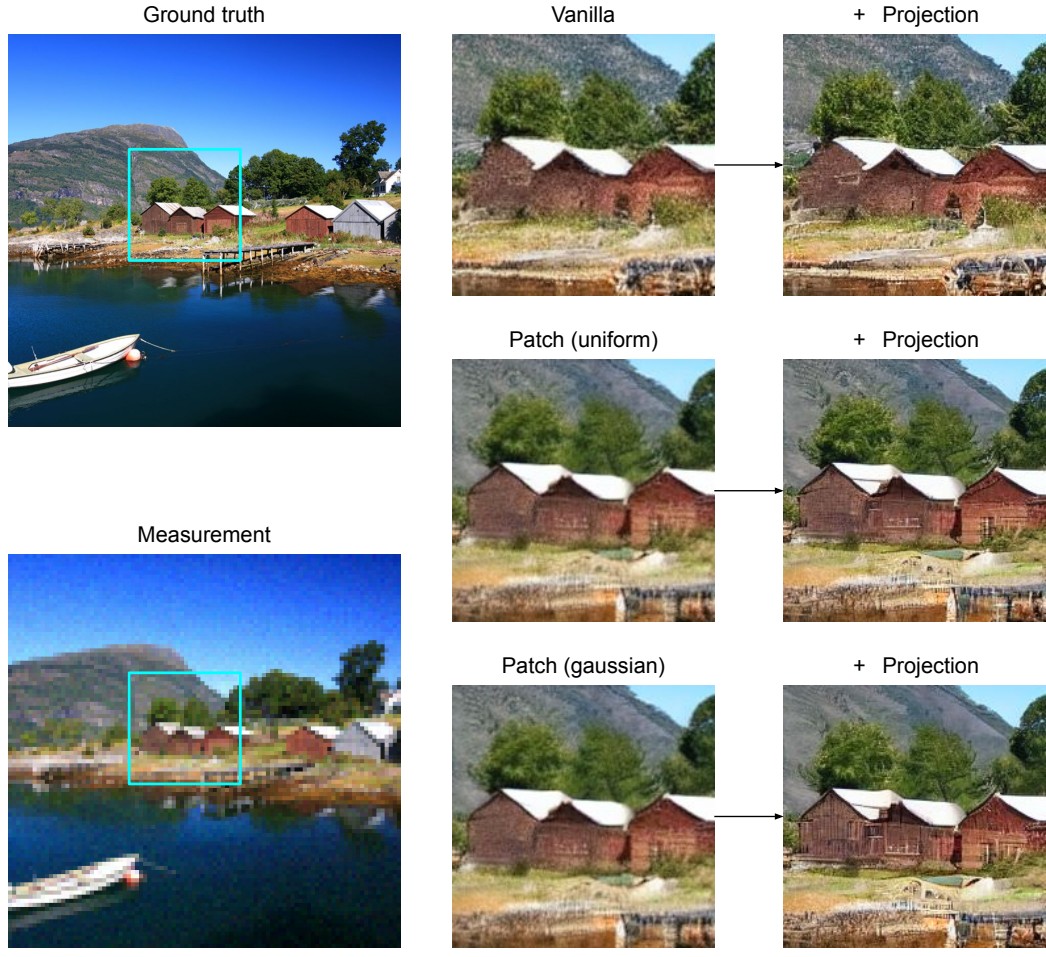

Figure 6: Further results on ×8 SR on DIV2K validation set of 768×768 resolution. Comparison between with and without using our projection approach on various baseline methods.

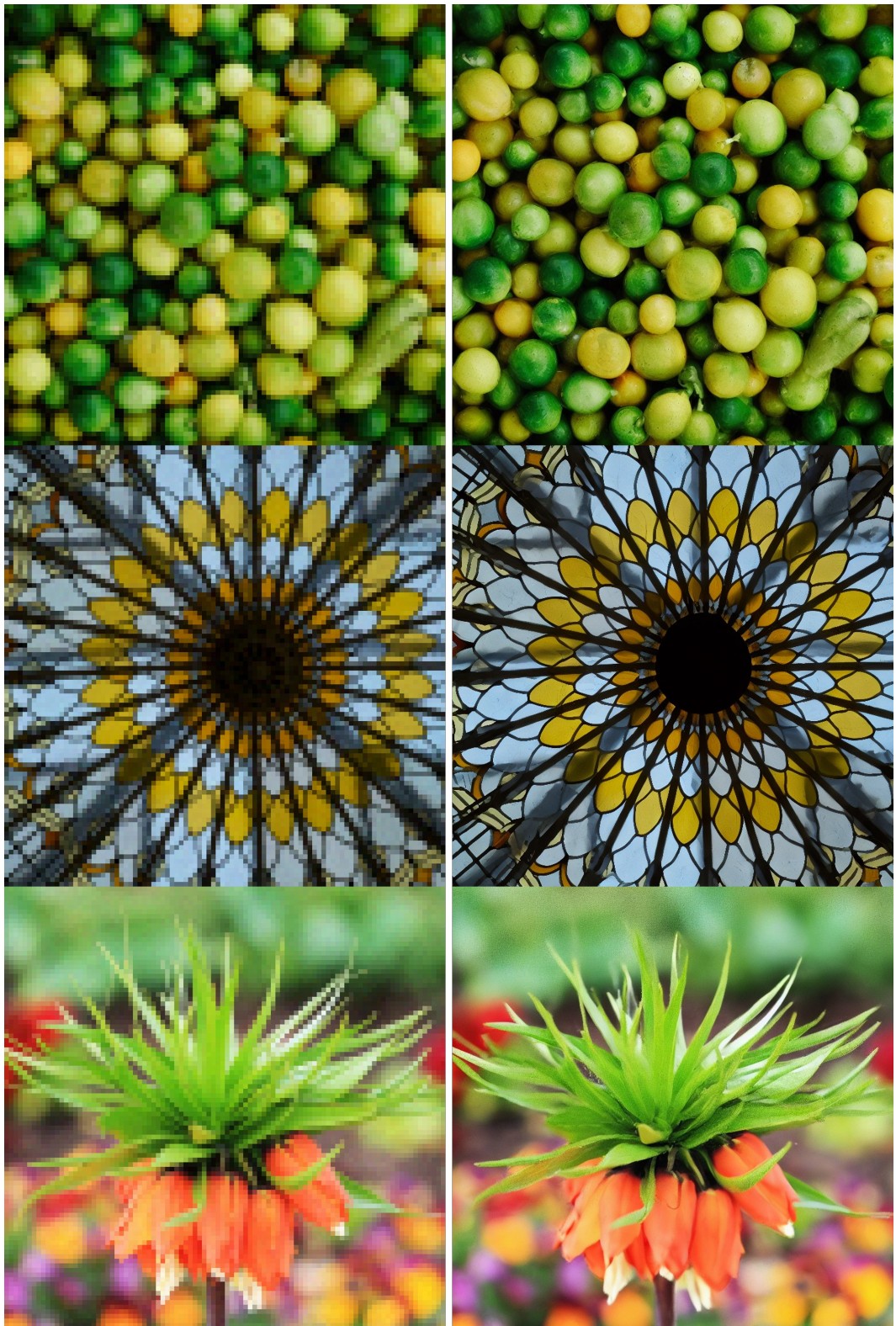

Figure 7: Full image results of ×8 SR on DIV2K validation set of 768×768 resolution. Left: measurement, Right: P2L

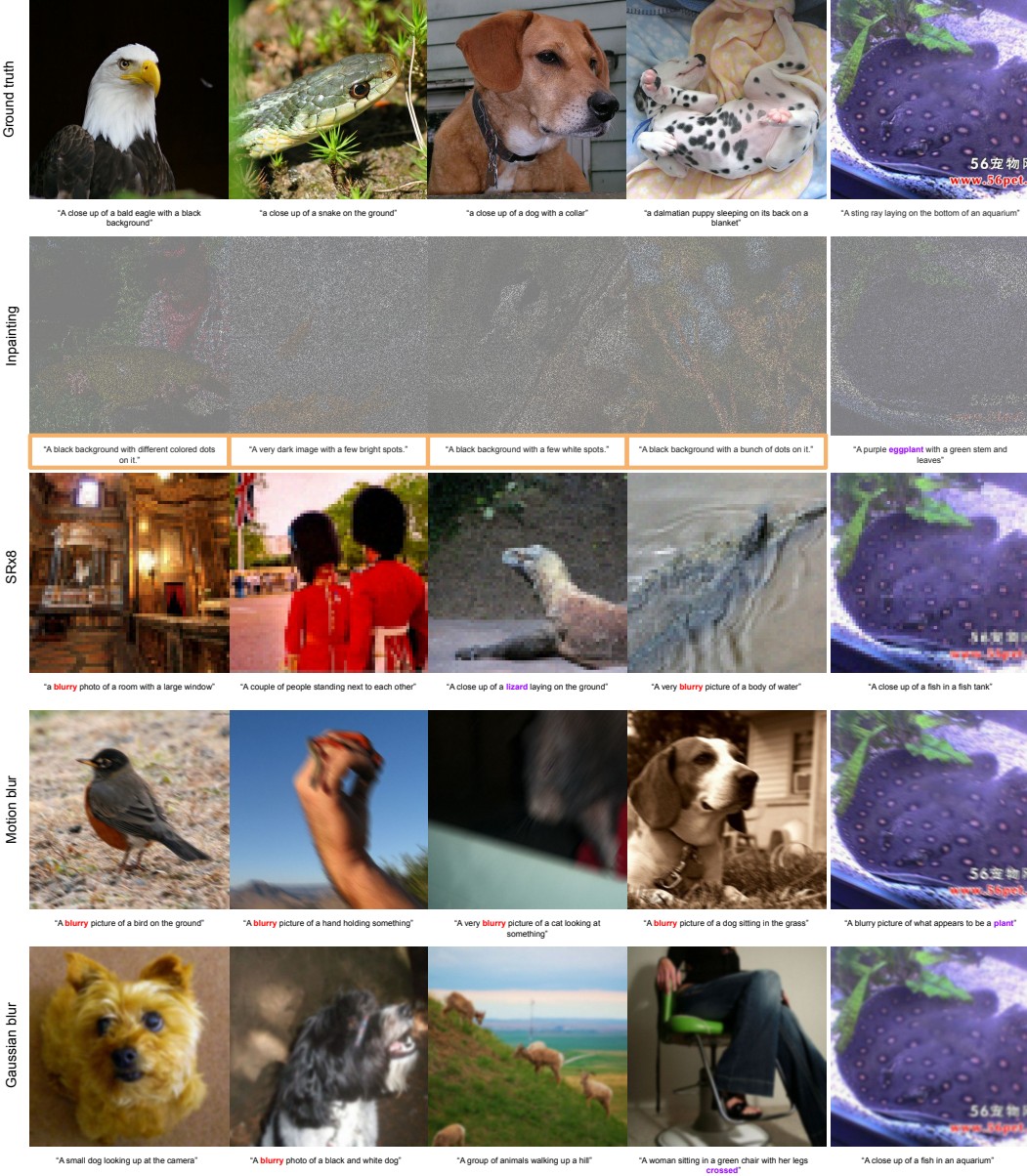

Figure 8: Captions generated by PALI (Chen et al., 2022) from ground-truth ImageNet 512×512 clean images, and the degraded images. The rightmost column contain images that are from the same ground truth. Captions in in orange box completely fail to describe the underlying image. Purple captions wrongly identify the image. Captions generated from degraded measurements often contain negative words such as blurry.

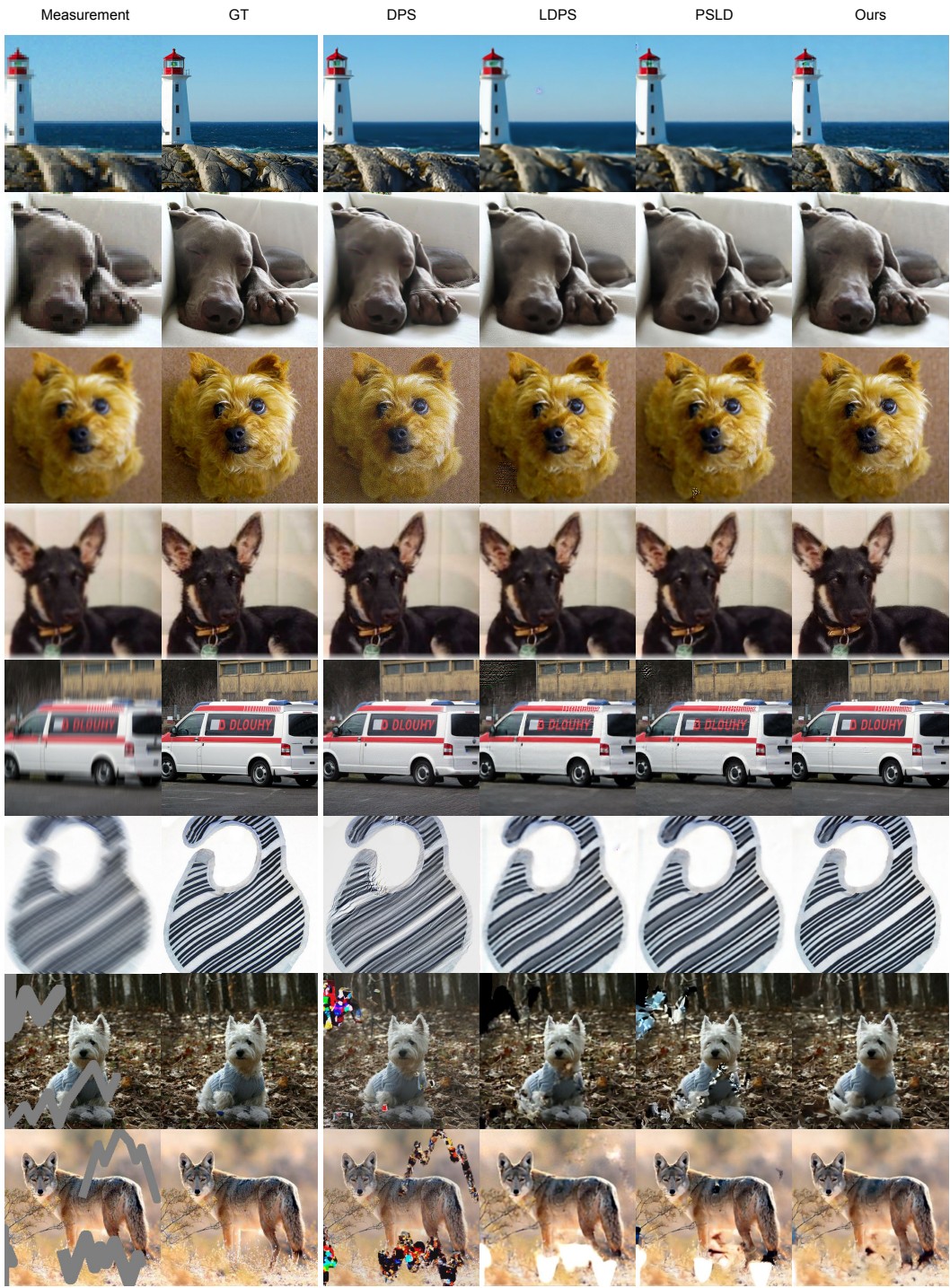

Figure 9: ImageNet restoration results. Row 1-2: SR×8, row 3-4: gaussian deblurring, row 5-6: motion deblurring, row 7-8: freeform inpainting; All with $\sigma = 0.01$ noise.

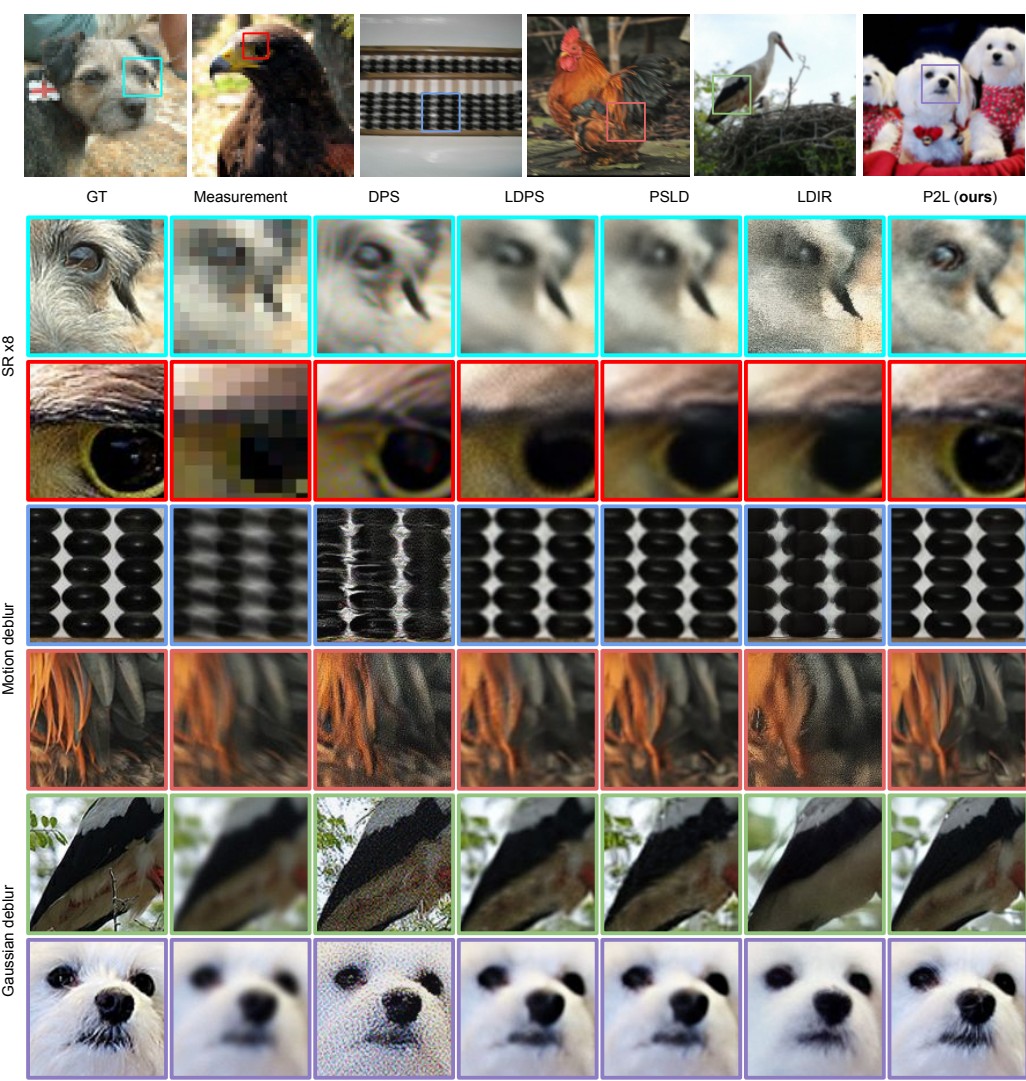

Figure 10: Close-up comparison on diverse inverse problem tasks. Ground truth, measurement, DPS (Chung et al., 2023b), LDPS, PSLD (Rout et al., 2023), LDIR (He et al., 2023), and the proposed method.

