# OpenReview forum: "Prompt-tuning Latent Diffusion Models for Inverse Problems"
_ICLR.cc/2024/Conference — Submitted to ICLR 2024_

### Official Review · Reviewer_QCsq · 2023-10-25

**Soundness:** 3 good
**Presentation:** 3 good
**Contribution:** 3 good
**Rating:** 6
**Confidence:** 5

**Summary:**

This paper aims at solving image inverse problems using latent diffusion models. Two main contributions are included. The first is that the authors introduce a method for prompt tuning, i.e., a way to automatically find the right prompt to condition diffusion models when solving inverse problems. The second is that the authors propose a method to keep the evolution of latent variables within the range space of the encoder, by projection optimization. According to the presented experiment, the proposed method seems to outperform existing state-of-the-art.

**Strengths:**

The idea of automatic prompt-tuning is novel and effective. The overall performance achieves state-of-the-art in solving image inverse problems.

**Weaknesses:**

The whole pipeline is somewhat complicated, involving many pyper-parameters which may be difficult to tune, as shown in Table 6.

The proposed method seems to be expensive in computation. Reports on the inference time are necessary.

Although I believe that the proposed method can solve large-scale inverse problems, there is no experimental evidence to prove that. It is recommended that the appendix provide more experimental results with general large images.

The authors mentioned range space projection, which is highly related to DDNM [1] and should be discussed.


Reference: [1] Wang et al., zero-shot image restoration using denoising diffusion null-space model, ICLR 2023.

**Questions:**

Please see the Weaknesses.

---

> ### Author Response · Authors · 2023-11-19
> **Reply to reviewer QCsq**
>
> Thank you for your thoughtful and constructive review.
>
> **W1. Method is complicated. There are many hyper-parameters to tune**
>
> **A.** We agree that there are many hyper-parameters involved in P2L. However, we would like to note that the different choices made for specific inverse problems are merely to achieve the best performance, and not because they are necessary. We can still achieve improved performance as opposed to baselines, which are evident in Table 1 and 3, where we use standard GD with 1.0 step size.
>
> **W2. Method is expensive in computation. Reports on the inference time are necessary**
>
> **A.** Thanks for pointing this out. We have now included inference times in Appendix C, Table 5, and included a discussion on the main text of Section 4. Please see general comment C2. We would like to 4. while solving inverse problems can lead to a significant improve in quality. We believe that future investigations will yield methods that can build upon P2L to achieve solvers that are faster.
>
> **W3. There is no evidence that the proposed method can solve large-scale inverse problems**
>
> **A.** We agree that including full images rather than patches of results in Figure 3 would be beneficial. In Appendix G, we included some example results.
>
> **W4. Range space projection is highly related to DDNM [1]**
>
> **A.** The range space discussed in this paper is the range space of the VAE encoder, which is irrelevant from the range space of the forward imaging operator that DDNM considers. We have clarified this in the revised version. In particular, for understanding of whole pipeline, we have derived a new theoretical justification of each step of the algorithm using an alternating minimization perspective. We believe that this derivation provides a theoretical justification of the projection step and help readers to understand the difference from DDNM.
>
>
> **References**
>
> [1] Wang et al. “Zero-shot image restoration using denoising diffusion null-space model” ICLR 2023.

---

> > ### Comment · Reviewer_QCsq · 2023-11-21
> >
> > The author's rebuttal addresses most of my concerns, and I'll keep my score.

---

### Official Review · Reviewer_fyie · 2023-10-30

**Soundness:** 2 fair
**Presentation:** 2 fair
**Contribution:** 2 fair
**Rating:** 3
**Confidence:** 4

**Summary:**

This paper proposes to use text prompt with iterative optimization for solving imaging inverse problems and constrains the evolution of latent variables using projection. Experimental results on high-resolution inverse problems show that the proposed method outperforms several DIS and LIDS methods on a variety of tasks.

**Strengths:**

1.  The motivation of using learnable prompt to improve the performance is meaningful.
2.  Experiments on different kinds of image inverse tasks including super-resolution, deblurring, and inpainting are performed.

**Weaknesses:**

1.  The writing is poor and the submission is hard to follow.

2.  Why we need iterative optimization similar to EM algorithm for optimizing text prompt and latent variable? The submission lacks necessary theoretical analysis and experimental evaluation.

3.  The proposed projection is similar to that proposed in Chung et al. (2023b). Why the proposed approach can provide a good regularization is not clearly elaborated and why this approach is named projection is not clear.

4.  No further analysis on why the proposed projection can target any resolution and only subjective descriptions are proposed.

5. The practicality of the proposed method is limited. The proposed P2L appears to be significantly slow. It necessitates 1000 reverse diffusion steps to achieve the results presented in the paper, each consisting of K backward propagations (BP) through the LDM for prompt tuning and 1 BP for a DPS-like step. It means that P2L could potentially be K times slower than DPS.

6. The lack of experiments on the commonly used datasets of size 256x256 provides less evidence to support that the proposed prompt tuning is useful. The performance of the compared baselines on datasets of size 512x512 reported in this paper is unreliable. For example, there is a substantial degradation in performance for DIS methods, which may attribute to the high sensitivity to hyperparameters of these methods. I suggest the authors to add experiments on datasets of size 256x256 for fair comparison.

7. Comparison with the DIS method termed PGDM [R1] should be included. It does not require extensive hyperparameter tuning. Additional comparison with it on datasets of size 512x512 would be beneficial to demonstrate the strength of the proposed method.
[R1] Song, Jiaming, et al. "Pseudoinverse-guided diffusion models for inverse problems." International Conference on Learning Representations. 2022.

-------------------------------------------------
**Post-Rebuttal Comments**

Part of my issues have been addressed, and I appreciate the authors' efforts to reorganize and rewrite the paper. However, I still have some concerns on the paper.

Although the authors have made great effort to rewrite the derivations for the method, I still find the writing in the revised version is hard to follow and confusing. Specifically, my confusions are listed as below:

1. From Eq. (6) to Eq. (7), the proposed method can be interpreted as a maximum a posterior based method. However, Eq. (8) implies that the method essentially becomes a posterior sampling based method. Besides, in principle, since the text prompt can be only inferred from the measurement, there are no additional information introduced for improving performance. I feel like the proposed method aims to sample from p(x,c|y) rather than directly sample from p(x|y).

2. I am confused with DDS approximation in Eq. (38). How does Eq. (38) relate to the original form of DDS approximation described in Eq. (48)? I do not find details or justification of DDS approximation.

Another concern is that the authors believe that P2L can act as a widely-used baseline as DPS. However, P2L is complex, much slower, and requires extensive hyper-parameter tuning than previous methods, which is contrary to the simple implementation of DPS. Therefore, it is doubtful to me that P2L could be an impactful baseline as DPS. Despite that the idea of optimizing text prompt is interesting, the realization of the idea is far from satisfactory.

Under these considerations, I decide to keep my score.

**Questions:**

Please refer to the Weaknesses.

---

> ### Author Response · Authors · 2023-11-19
> **Reply to reviewer fyie (1/2)**
>
> Thank you for your thorough constructive review. Please see our response below.
>
> **W1. The writing is poor and the submission is hard to follow.**
>
> **A.** The main focus of this work was to propose a first method that leverages text-prompt embedding for solving inverse problems using latent diffusion models. Moreover, we have improved our manuscript also in the theoretical side from an alternating minimization perspective for a better understanding of why our method works better, and on which component our algorithm is improving the previous solvers.
>
> 1. Optimizing the text embedding can be thought of as corrections that are made on the posterior PF-ODE trajectory of reverse diffusion that modifies the embedding to meet the measurement conditions. Under this view, we do not need any assumptions on the independence between the noisy latents and the embedding.
>
> 2. The Encoder range space projection can be understood as performing MAP optimization under the VAE prior. Per this view, P2L can be seen as the first to leverage both the diffusion and the VAE generative prior into the posterior sampling scheme, further highlighting the strength of the approach.
>
> These modifications offer better understanding on how the method can be derived and understood, and what type of approximation is used, and similarity and differences from the existing LDIS approaches such as PSLD. In this paper, the focus is more on the proof of concept that prompt tuning can yield improved performance, and that we can achieve better results by incorporating the prior learned through the VAE.
>
> We have updated Section 3. We would appreciate it if the reviewer can point us to any specific section that the reviewers found it unclear or hard to follow.
>
> **W2.Why do we need iterative optimization similar to EM?**
>
> **A.** As discussed in General Comment C1, in this revision we have provided a unified theoretical perspective using alternating minimization.  Moreover, Eq. (6) and (7) justify the need for prompt optimization.
>
> More specifically, if we don’t optimize for the text, we have subpar performance, which is the main point that we want to make in this paper: prior approaches only optimize for the latent variable. If we do not optimize the latent variables, then we cannot find a solution compatible with the measurements. This is why the optimal performance is obtained when simultaneously or alternatively optimize both text and latent representations.
>
> As stated in the answer to weakness 1, we have made our best efforts to clarify the theoretical implications of the proposed work. The optimization of text-prompt embedding is required as it is an unknown variable that is used throughout the reverse diffusion PF-ODE trajectory. The remaining sampling algorithm that synergistically combines optimization in both the pixel- and the latent-space can be regarded as MAP optimization of a specific loss function by splitting the variables.
>
> **W3. The proposed projection is similar to DPS [1]. Why the method is named projection is unclear**
>
> **A.** The proposed projection is completely irrelevant to DPS [1]. Note that in [1] there is no VAE or latent diffusion model, nor no exploitation of the text prompt. The two main contributions of this work are: 1. Exploit the text dependency in inverse problems by jointly optimizing the text representation; 2. Exploit the VAE prior to making the framework work on high-dimensions (through better latent diffusion models).
>
> Regarding the specific proposed projection using the VAE prior, we elaborated on why the proposed projection approach yields good regularization in Section 3.2. The proposed method is called projection, as it clearly projects the latents onto the range space of the encoder.
>
>
> **References**
>
> [1] Chung et al. “Diffusion posterior sampling for general noisy inverse problems”, ICLR 2023
>
> [2] Rout et al. “Solving linear inverse problems provably with latent diffusion models”, NeurIPS 2023
>
> [3] He, Linchao, et al. "Iterative reconstruction based on latent diffusion model for sparse data reconstruction." arXiv preprint arXiv:2307.12070 (2023).
>
> [4] Song, Bowen, et al. "Solving inverse problems with latent diffusion models via hard data consistency." arXiv preprint arXiv:2307.08123 (2023).
>
> [5] Song, et al. "Pseudoinverse-guided diffusion models for inverse problems." ICLR 2023

---

> ### Author Response · Authors · 2023-11-19
> **Reply to reviewer fyie (2/2)**
>
> **W4. No further analysis on why the proposed projection can target any resolution and only subjective descriptions are proposed.**
>
> **A.** Thanks for pointing this out. We have modified Appendix F to fully elaborate on the analysis. The proposed approach is useful for high-resolution image restoration because we are fully leveraging the power of the pre-trained VAE, and because these VAEs are much more robust to the change in the input resolution. In our method, we are proposing a method that can fully utilize this property, by alternatively optimizing latent and pixel space (through the VAE). In the revised version we have incorporated a theoretical analysis where we leverage the VAE prior within the formulation (now pointed out in Section 3.2).
>
> **W5. The practicality of the proposed method is limited as it is slow.**
>
> **A.** Please see general comment C2. We agree that P2L requires many iterations. However, we would like to emphasize that P2L is more a proof of concept that optimizing text-prompt embeddings while solving inverse problems can lead to significantly better results. We believe that future investigations will yield methods that can build upon P2L to achieve solvers that are faster. Also, we would like to mention that there are certain use cases that do not require a fast solver but they require to reach the best possible quality (For example, one could use a "slow" method to generate the best possible result on on real data – where we don't know the ground-truth or reference, and later train a supervised method using the paired data).
>
> **W6. Lack of experiments on 256$\times$256 data provides less evidence. Experiments on 512$\times$512 is unreliable. Degradation in performance of DIS could be due to sensitivity of hyperparameters.**
>
> **A.** We respectfully disagree. When running all DIS baselines, we did careful hyper-parameter tuning to yield the best possible result out of all methods. The reason for DIS underperforming on 512$\times$512 resolution is because pixel diffusion models that directly target this resolution are often low quality. This is why latent diffusion models and cascaded models dominate the mainstream when targeting image sizes beyond 256$\times$256. In this work, we specifically focus on solving inverse problems on higher resolutions beyond 256$\times$256, which is why we proceed on the latent space (i.e., Stable Diffusion).
>
> In all prior works that tried to leverage latent diffusion models as priors for solving inverse problems [2,3,4], some experiments involve newly training a VAE+LDM from scratch for 256$\times$256 resolution, specified for a dataset that they are considering. This is not what we aim for, as explicitly pointed out in the last paragraph of Section 1: we are interested in using a foundational generative model that can operate on a fully general domain.
>
> **W7. Comparison with $\Pi$GDM [5]**
>
> **A.** Please see modified Table 2, 8. We additionally compare against $\Pi$GDM [5].
>
> **References**
>
> [1] Chung et al. “Diffusion posterior sampling for general noisy inverse problems”, ICLR 2023
>
> [2] Rout et al. “Solving linear inverse problems provably with latent diffusion models”, NeurIPS 2023
>
> [3] He, Linchao, et al. "Iterative reconstruction based on latent diffusion model for sparse data reconstruction." arXiv preprint arXiv:2307.12070 (2023).
>
> [4] Song, Bowen, et al. "Solving inverse problems with latent diffusion models via hard data consistency." arXiv preprint arXiv:2307.08123 (2023).
>
> [5] Song, et al. "Pseudoinverse-guided diffusion models for inverse problems." ICLR 2023

---

> ### Author Response · Authors · 2023-11-21
> **Nearing the end of the discussion period**
>
> Dear reviewer fyie,
>
> As the deadline for the reviewer-author discussion phase is fast approaching (there is only a day left), we respectfully ask whether we have addressed your questions and concerns adequately. We would be happy to clear up any additional questions.

---

> ### Author Response · Authors · 2023-11-22
> **[Reminder] Summarization of our rebuttal**
>
> Dear reviewer fyie,
>
> We believe that we have addressed the concerns that you have raised. Specifically,
>
> 1. We have **clarified the theory** of P2L by deriving the algorithm through the lens of **alternating minimization**. This also highlights the need for iterative optimization needed for optimizing each component.
>
> 2. We further analyzed on why P2L can solve arbitrary-resolution image restoration problems and included full-resolution results.
>
> 3. We made sure that the **runtime of P2L is discussed and compared against the baselines**. We also highlighted the impact of P2L as the **first proof-of-concept** and for use in cases where speed is not the primary issue.
>
> 4. We included **additional comparisons** against the baseline that the reviewer suggested and clarified why the experiments were conducted on 512x512 resolution.
>
> We would like to gently remind you that the **end of the discussion period is imminent**. We would appreciate it if you could let us know whether our comments addressed your concerns.
>
> Best regards,
> Authors

---

### Official Review · Reviewer_vCQL · 2023-11-01

**Soundness:** 3 good
**Presentation:** 3 good
**Contribution:** 3 good
**Rating:** 6
**Confidence:** 4

**Summary:**

This paper focuses on solving inverse problems using a new method based on (latent) diffusion models. Prior works using similar generative priors only update the latent variables while keeping the text-prompt null in most cases. The authors motivate their approach by showing experiments where better text-conditioning helps achieve a significant gain in performance. Finally,  the authors propose a method that optimizes the null-embedding along with latent variables to minimize the measurement error. It extends the prior works done by Rout et al. and He et al. to optimizing the null-embeddings in addition to the latents. In addition, the authors propose a projection step to ensure that the generated image resides in the range space of the decoder. Extensive experiments on FFHQ and ImageNet are conducted to support the claims.

**Strengths:**

1. Optimizing the null-embeddings in addition to the latents is a strong contribution and very useful in several downstream applications.
2. The authors achieve state-of-the-art performance in several tasks.
3. The paper is well-written and the main points are clearly discussed with sufficient details to reproduce the results.

**Weaknesses:**

1. In Section 3.2, the authors conduct an experiment using PSLD to show that it always diverges even if it started from a clean image. I believe that this experiment does not offer any insights because of two reasons:
(i) The approximation used is not what was proposed in PSLD. In fact, the authors of PSLD show that aiming for any fixed point is not a good idea. Instead, they prove that the gluing objective helps recover the unique fixed point that exhibits contraction towards the optimal solution in a linear setting (Theorem 3.6).
(ii) The fixed point solver works only for a unique step size (Theorem 3.5). Therefore, it is hard to to get any meaningful conclusion from this experiment unless the step size is properly tuned or gluing objective is used instead.

Also, what task was considered in this experiment?

2. An important point to discuss is that the gluing objective is in fact projecting onto the measurements. But it uses the analytical solution and does one step of gradient descent to save computational time. So the trade-off between performance and speed should be discussed in the proposed projection optimization step.

3. Choice of $\tau$: It is obvious that multiple steps of gradient descent would lead to better results than a single step. To make a fair comparison, I believe the running time should be compared. In addition, there is another optimization problem being solved at every step of diffusion to get the near-optimal embeddings.

4. The description about PSLD in the paragraph below eq. (6) is not fully correct. PSLD shows that there exists a step size for which guiding towards the fixed point is a good idea. Not for any step size.

5. Analysis of the proposed algorithm in a  simple linear model setting might be helpful to better understand the contributions in terms of convergence guarantees.

**Questions:**

1. How do you go from equation (11) to (12)? It is assumed that $C$ and $z_t$ are independent. However, this assumption is not properly justified?
2. Footnote 3 on page 5 is not the finding of this paper. It was shown in prior works (e.g. PSLD) that naively scaling DPS does not work.
3. How many $C$ updates per $z_t$ update is needed? How does that affect the number of neural function evaluations? This  should be discussed properly in the main draft.
4. The use of some params like K is not clearly shown in Algorithm 2. It should be used as an argument to the optimization block.

-----------------------------------
## After discussion:
I thank the authors for their response. The idea of tuning the prompts used in SoTA inverse problem solvers is an important contribution. However, the proposed algorithm currently takes around **30 mins** to process **one single image** as opposed **5 mins for DPS** and **7 mins for PSLD/GML-DPS**. I understand that the current work is at its early stage and its runtime might improve in the near future. I believe the current score "marginally above acceptance threshold (6/10)" is a good assessment of the proposed method. Therefore, I am planning to keep my score.

---

> ### Author Response · Authors · 2023-11-19
> **Reply to reviewer vCQL**
>
> Thank you for your positive and constructive comments. Please see our response below.
>
> **W1. Relation to PSLD [1]**
>
> **A.** Thank you for the thoughtful comment! We would like to clarify the experiment conducted in section 3.2. in the original submission. In this experiment, there is no *diffusion* component. We only iterate the encoding-decoding using the VAE used in Stable Diffusion v1.4. Hence, the experiment is *not* showing that PSLD always diverge. In fact, there is no inverse problem here, just the repeated application of the VAE encoder and decoder. Even when additionally using the gluing objective in between the encoding-decoding procedure, we still found that the encoding-decoding does not provide a fixed point solution. That being said, we also agree with the reviewer’s point on potential miscommunication to the readers, so we have removed the section in the revised version.
>
> **W2. Gluing projects onto measurements. Trade-off between gluing and proximal optimization?**
>
> **A.** We agree that is an important issue. Table 4 of the modified manuscript discusses that the gluing objective is indeed fast projecting to the measurements. As mentioned in the paper, the proposed proximal optimization step can be done with negligible computation overhead, nearly similar to that of the projection. We newly include the comparison of computation cost in the manuscript. Please refer to the modified final paragraph of Section 4.
>
> **W3. Choice of $\tau$? Text-prompt embedding is solved by multiple-step optimization**
>
> **A.** Thank you for pointing this out. Indeed, in prompt embedding optimization, we are solving an optimization problem rather than taking a single gradient step. We modified section 3.1. to give a better explanation about the method. Please consult general comment C1. Also, we include the comparison in the compute time for each algorithm in our manuscript (Table 5).
>
> **W4. Imprecise description about PSLD [1]**
>
> **A.** Thanks for pointing this out. We have modified the description to be more precise.
>
> **W5. Analysis of the proposed algorithm missing**
>
> **A.** Thanks for the constructive comments. As discussed fully in General Comment C1, rather than using linear model, we have now derived each step of the algorithm from an alternating minimization perspective using VAE prior. This derivation clearly explains why each step of the algorithm is necessary and what type of approximation is required to arrive at the simplified solution.  Although we could not provide an exact convergence analysis due to the associated approximation at each step, we believe that the unified framework explains the similarity and difference between our method and existing approaches such as PSLD.
>
> **Q1. Assumption on independence**
>
> **A.** In the revised version we have removed this assumption. Please see general comment C1 and modified section 3.1.
>
> **Q2. Footnote 3 is not the finding of this paper**
>
> **A.** The footnote was used for emphasizing that using the DPS for pixel- and latent- diffusion models lead to different consequences, and not to propose a new idea. That said, we agree that this may be redundant. The footnote was removed.
>
> **Q3. How many $\mathcal{C}$ per $\mathbf{z}_t$ update? Runtime analysis needed**
>
> **A.** Please see general comment 2. We now discuss the compute time that scales with the number of update steps used for $\mathcal{C}$ in the main paper. Please see Table 5.
>
> **Q4. $K$ should be inside the optimization block**
>
> **A.** Corrected. We appreciate the careful reading!
>
>
>
> **References**
>
> [1] Rout et al. “Solving linear inverse problems provably with latent diffusion models”, NeurIPS 2023

---

### Official Review · Reviewer_7Ao9 · 2023-11-10

**Soundness:** 3 good
**Presentation:** 4 excellent
**Contribution:** 3 good
**Rating:** 6
**Confidence:** 5

**Summary:**

This paper proposed a method (P2L) for prompt-tuning latent diffusion models to solve inverse problems. The method jointly optimises the latent variables/prompts on-the-fly and reconstructs the image through the inverse diffusion process. The prompt tuning step helps to tailor the model to the specific task of solving the inverse problem, while the projection step helps to ensure that the generated images are realistic and free of artefacts. The experiments on super-resolution, deblurring and inpainting demonstrate the effectiveness of P2L.

**Strengths:**

(1) The idea of learning prompts to guide the diffusion models for inverse problems is very interesting.

(2) The method is technically sound.

**Weaknesses:**

The paper lacks a theoretical analysis of, for example, convergence.

The results are not very promising.
(1) From Table 1, the performance (PSNR) gains of P2L are subtle.
(2) From the ablation experiments in Table 4, the difference between the results obtained by not using any of the three proposed modules and the results obtained by using all of them is not significant.
(3) From Table 5, the proposed proximal calibration is not that superior to the projection-based calibration, which is even cheaper and faster.

**Questions:**

(1) I'm curious if you have tried to figure out the text corresponding to the learned prompt. As shown in Table 1, including the methods presented in this paper and PALI's for prompt generation, what kind of textual information do they represent? I am curious about the interpretability of the prompts.

This may be important for e.g. medical image reconstruction, or at least it is unclear to me whether there are reasonable (textual) prompts to guide the reconstruction of medical images.

(2) In algorithms 1-4 there are two functions called 'OptimizeEMB', which 'OptimizeEMB' do you use in algorithm 4?

---

> ### Author Response · Authors · 2023-11-19
> **Reply to reviewer 7Ao9**
>
> Thank you for your positive and thoughtful comments. Please see our response below.
>
> **W1. The paper lacks a theoretical analysis of, for example, convergence.**
>
> **A.** Please see our general response C1. While we limit the scope of our work and do not consider analysis of convergence, we have significantly extended our explanations on the theoretical motivations of P2L using an alternating minimization framework and clarified the approximation at each step. We also clarified the link to the existing LDIS approach such as PSLD. We believe that this extension adds much value to our manuscript, and we hope the reviewer also finds this interesting.
>
> **W2. The results are not very promising.**
>
> **A.** We would like to respectfully disagree that the improvements are subtle. It is widely known that PSNR (distortion) and FID (perception) are two metrics that have trade-off [1,2]. When one opts for better PSNR, it leads to worse FID and vice versa. In this regard, the fact that our method keeps the PSNR level and improves FID by a large margin shows that P2L is actually pushing the pareto frontier forward.
>
> 1. The performance gains are mostly seen in the FID score: *perceptual quality*. The information from the measurement may be sufficient for minimizing the *distortion*, but insufficient to achieve the best *perceptual* quality. The text embedding that we optimize for fills in this gap, as can be clearly seen in **-2.4, -0.9, -11.7,-4.0** improvements in the FID score.
> 2. The take-home message from Table 4 (now Table 3) is that the proposed components in the paper are *synergistic and provides a better trade-off*, and do not hamper each other. Moreover, similar to (1), the perceptual quality measured by FID does significantly increase.
> 3. The compute time required for latent diffusion model-based inverse problem solvers including P2L and all the baselines are almost perfectly linear on NFE. The difference between the compute time for using projection and proximal optimization is marginal. Moreover, while the difference between gluing and our proximal calibration may be not as pronounced on smaller noise levels, it gets more significant as we increase the noise level.
>
> **Q1. Text corresponding to learned prompt?**
>
> **A.** Thank you for the thoughtful comment. Indeed, it would be very interesting and helpful if we could find out the “learned prompts” after the optimization. However, this is not possible as we leverage CLIP encoder, which does not have a corresponding decoding architecture. That said, using a diffusion model that leverages e.g. T-5 encoder which has a corresponding decoder, we believe that the learned prompts would be more explainable, which we leave as a future direction of research.
>
> **Q2. Which 'OptimizeEMB' do you use in algorithm 4?**
>
> **A.** The ‘OptimizeEMB’ function is defined in a separate algorithm. In order to clarify this, we include Algorithm 2 in the appendix which corresponds to the definition of the ‘OptimizeEMB’ function with annotations. The same function is used for both Algorithm 1 and 3, which are the vanilla, and the adam versions of P2L.
>
>
>
>
> **References**
>
> [1] Blau, Yochai, and Tomer Michaeli. "The perception-distortion tradeoff." CVPR 2018.
>
> [2] Delbracio, Mauricio, and Peyman Milanfar. "Inversion by direct iteration: An alternative to denoising diffusion for image restoration." TMLR (2023).

---

### Author Response · Authors · 2023-11-19
**General response to reviewers**

We would like to thank the reviewers for their constructive and thorough reviews. We are encouraged that the reviewers think that **the idea is very interesting and method technically sound** (7Ao9), **makes a strong contribution, very useful, achieves SOTA, well-written** (vCQL), **the motivation is meaningful** (fyie), and **the idea is novel and effective, achieving SOTA performance** (QCsq).

There were two major comments that were overlapping among the reviewers:

**C1. The paper lacks theoretical analysis (e.g. convergence). Some of the derivations that lead to the method are not justified.**

**A.** The main focus of this work was to propose a first method that leverages text-prompt embedding for solving inverse problems using latent diffusion models. While we do not fully analyze the convergence properties of P2L, we would like to emphasize that we are proposing the first step towards using text-prompts in the field of inverse problem solving. Having said that nonetheless, we have improved our manuscript also in the theoretical side for a better understanding of why our method works better, and on which component our algorithm is improving the previous solvers.

1. Optimizing the text embedding can be thought of as corrections that are made on the posterior PF-ODE trajectory of reverse diffusion that modifies the embedding to meet the measurement conditions. Under this view, we do not need any assumptions on the independence between the noisy latents and the embedding.
2. The Encoder range space projection can be understood as performing MAP optimization under the VAE prior. Per this view, P2L can be seen as the first approach to leverage both the diffusion and the VAE generative prior into the posterior sampling scheme, further highlighting the strength of the approach.

Summing up, the optimization of text-prompt embedding is required as it is an unknown variable that is used throughout the reverse diffusion PF-ODE trajectory. The remaining sampling algorithm that synergistically combines optimization in both the pixel- and the latent-space can be regarded as MAP optimization of a specific loss function by splitting the variables. We have now derived each optimization step as an alternating minimization, and clarified what type of approximation is used in each step. The derivation also clearly explains the link to the existing LDIS approach such as PSLD.

We believe that these modifications offer better understanding on how the method can be derived and understood. In this paper, the focus is more on the proof of concept that prompt tuning can yield improved performance, and that we can achieve better results by incorporating the prior learnt through the VAE.

**C2. Running time should be measured and discussed. The method is slow.**

**A.** The running time and the comparison against other baselines are now included in the manuscript. As the reviewers mentioned, P2L requires more compute against other LDIS baselines as we additionally optimize for the text prompt. However, it should be noted that P2L is the first approach that shows the possibility and feasibility of the approach. While it may not be computationally efficient at this point, we strongly believe that P2L would be a good cornerstone that future works can build upon to devise faster, more efficient solvers.

We would like to shed light on DPS [1], which, when first developed, was, and still is, one of the slowest diffusion model-based inverse problem solvers at the time. Yet, the high performance and wide applicability led to the method being one of the widely-used, compared-against baseline which others build upon to yield more efficient solutions. P2L, in our humble opinion, will have an analogous influence on the literature regarding leveraging prompts in LDIS.

Please also refer to our point-to-point response provided as reply to the respective comments.

**References**

[1] Chung et al. “Diffusion posterior sampling for general noisy inverse problems”, ICLR 2023

---

### Meta-Review · Area_Chair_yGaH · 2023-12-10

**Metareview:**

The authors propose using text-to-image latent diffusion models as general priors for solving imaging inverse problems such as super-resolution, deblurring, and inpainting. They propose a method for prompt tuning that jointly optimizes the text embedding on-the-fly while running the reverse diffusion process and a projection step to ensure that the generated images are realistic.

- Reviewer 7Ao9 finds that "the paper lacks a theoretical analysis of, for example, convergence" and that "the results are not very promising".
- Reviewer vCQL points out that "in section 3.2 an experiment using PSLD always diverges" due to incorrect settings, that "the trade-off between performance and speed should be discussed", need for "fair comparison" and for "analysis" and convergence guarantees.
- Reviewer fyie thinks that "writing is poor and the submission is hard to follow", "lacks necessary theoretical analysis and experimental evaluation", "projection similar to Chung et al. (2023b)", "The practicality of the proposed method is limited... P2L could potentially be K times slower than DPS", "Comparison with PGDM [R1] should be included"
- Reviewer QCsq considers that method is complicated with many hyper-parameters to tune, method is computationally expensive, inference times needed, and that there is no evidence that the proposed method can solve large-scale inverse problems, and range space projection is highly related to DDNM.

The authors provides responses to all the reviewers, however none of the reviewers is willing to champion the acceptance and one reviewer is firmly against acceptance (fyie). Reviewers expressed multiple concerns.

The meta-reviewer, after carefully checking the reviews, the discussions, and the paper, agrees that the paper requires a significant revision as it lacks in aspects pointed out by the reviewers, like novelty, clarity and details, and sufficient experiments and the method is computationally expensive.

The authors are invited to benefit from the received feedback and further improve their work.

**Justification For Why Not Higher Score:**

The paper does not meet the acceptance bar due to the multiple weaknesses and issues pointed out by the reviewers. A major revision and another review round is necessary.

**Justification For Why Not Lower Score:**

N/A

---

### Decision · Program_Chairs · 2024-01-16

Reject